# Glial Ca$^{2+}$ signaling links endocytosis to K$^+$ buffering around neuronal somas to regulate excitability

Shirley Weiss[1,2,3]*, Jan E Melom[1,2,3], Kiel G Ormerod[1,2,3], Yao V Zhang[1,2,3], J Troy Littleton[2]

[1]The Picower Institute for Learning and Memory, Massachusetts Institute of Technology, Cambridge, United States; [2]Department of Biology, Massachusetts Institute of Technology, Cambridge, United States; [3]Department of Brain and Cognitive Sciences, Massachusetts Institute of Technology, Cambridge, United States

**Abstract** Glial-neuronal signaling at synapses is widely studied, but how glia interact with neuronal somas to regulate their activity is unclear. *Drosophila* cortex glia are restricted to brain regions devoid of synapses, providing an opportunity to characterize interactions with neuronal somas. Mutations in the cortex glial *NCKX$^{zydeco}$* elevate basal Ca$^{2+}$, predisposing animals to seizure-like behavior. To determine how cortex glial Ca$^{2+}$ signaling controls neuronal excitability, we performed an in vivo modifier screen of the *NCKX$^{zydeco}$* seizure phenotype. We show that elevation of glial Ca$^{2+}$ causes hyperactivation of calcineurin-dependent endocytosis and accumulation of early endosomes. Knockdown of sandman, a K$_{2P}$ channel, recapitulates *NCKX$^{zydeco}$* seizures. Indeed, sandman expression on cortex glial membranes is substantially reduced in *NCKX$^{zydeco}$* mutants, indicating enhanced internalization of sandman predisposes animals to seizures. These data provide an unexpected link between glial Ca$^{2+}$ signaling and the well-known role of glia in K$^+$ buffering as a key mechanism for regulating neuronal excitability.
DOI: https://doi.org/10.7554/eLife.44186.001

*For correspondence:
s_weiss@MIT.EDU

Competing interests: The authors declare that no competing interests exist.

## Introduction

Glial cells are well known to play structural and supportive roles for their more electrically excitable neuronal counterparts. However, growing evidence indicates glial Ca$^{2+}$ signaling influences neuronal physiology on a rapid time scale. A single astrocytic glia contacts multiple neuronal cell bodies, hundreds of neuronal processes, and tens of thousands of synapses (*Halassa et al., 2007*; *Ventura and Harris, 1999*). Cultured astrocytes oscillate intracellular Ca$^{2+}$ spontaneously (*Takata and Hirase, 2008*) and in response to neurotransmitters (*Agulhon et al., 2008*; *Lee et al., 2010*), including glutamate (*Cornell-Bell et al., 1990*). Glutamate released during normal synaptic transmission is sufficient to induce astrocytic Ca$^{2+}$ oscillations (*Dani et al., 1992*; *Wang et al., 2006*), which trigger Ca$^{2+}$ elevation in co-cultured neurons (*Nedergaard, 1994*; *Parpura et al., 1994*) that can elicit action potentials (*Angulo et al., 2004*; *Fellin et al., 2006*; *Fellin et al., 2004*; *Pirttimaki et al., 2011*). These astrocyte-neuron interactions suggest abnormally elevated glial Ca$^{2+}$ might produce neuronal hypersynchrony. Indeed, increased glial activity is associated with abnormal neuronal excitability (*Wetherington et al., 2008*), and pathologic elevation of glial Ca$^{2+}$ can play an important role in the generation of seizures (*Gómez-Gonzalo et al., 2010*; *Tian et al., 2005*). However, the molecular pathway(s) by which glia-to-neuron communication alters neuronal excitability is poorly characterized. In addition, how glia interface with synaptic versus non-synaptic regions of neurons is unclear.

Several glia-neuronal cell body interactions have been reported for different glial subtypes (*Allen and Barres, 2009*; *Baalman et al., 2015*; *Battefeld et al., 2016*; *Takasaki et al., 2010*). A single mammalian astrocyte can be associated with multiple neuronal cell bodies and thousands of synapses (*Halassa et al., 2007*; *Ventura and Harris, 1999*). However, the complex structure of mammalian astrocytes and the diversity of their glia-neuron contacts makes it challenging to directly manipulate glial signaling only at contacts with neuronal cell bodies. *Drosophila* provides an ideal system to study glial-neuronal soma interactions as the *Drosophila* CNS contains two specialized astrocyte-like glial subtypes that interact specifically either with dendrites and synapses (astrocytes, *Stork et al., 2014*) or with neuronal cell bodies (cortex glia, *Awasaki et al., 2008*; *Pereanu et al., 2005*). Cortex glia encapsulate all neuronal cell bodies in the CNS with fine, lattice-like processes (*Awasaki et al., 2008*; *Coutinho-Budd et al., 2017*) (*Figure 1A*) and are thought to provide metabolic support and electrical isolation to their neuronal counterparts (*Buchanan and Benzer, 1993*; *Volkenhoff et al., 2015*).

Previous work in our lab identified zydeco (zyd), a cortex glial enriched $Na^+Ca^{2+}K^+$ (NCKX) exchanger involved in maintaining normal neural excitability (*Melom and Littleton, 2013*). Mutations in $NCKX^{zydeco}$ (hereafter referred to as *zyd*) predispose animals to temperature-sensitive seizure-like behavior (*Figure 1—video 1*, see Materials and methods) and result in bang sensitivity (seizure-like behavior following a brief vortex). Basal intracellular $Ca^{2+}$ levels are elevated in *zyd* cortex glia, while near-membrane microdomain $Ca^{2+}$ oscillations observed in wildtype cortex glia are abolished (*Figure 1—video 2* and *3*). Whether the loss of $Ca^{2+}$ microdomain events in *zyd* is due to a disruption in the mechanism generating these events or secondary to a saturation effect from elevated basal $Ca^{2+}$ levels is unclear. Though the mechanism(s) by which cortex glia modulate neuronal activity in *zyd* mutants is unknown, disruption of glial $Ca^{2+}$ regulation dramatically enhances seizure susceptibility.

To determine how altered cortex glial $Ca^{2+}$ signaling in *zyd* mutants regulates neuronal excitability, we took advantage of the *zyd* mutation and performed an RNAi screen for modifiers of the seizure-like phenotype. Here we show that chronic elevation of glial $Ca^{2+}$ causes hyperactivation of calcineurin-dependent endocytosis, leading to an endo-exocytosis imbalance. In addition, knockdown of sandman, a $K_{2P}$ channel, recapitulates the *zyd* phenotype and acts downstream of calcineurin in cortex glia, suggesting impaired sandman expression on cortex glial membranes is the cause of the *zyd* seizure phenotype. Indeed, cortex glial expression of GFP-tagged sandman shows that the protein is reduced on *zyd* cortex glial membranes. In addition, overexpressing a constitutively active $K^+$ channel in cortex glia can rescue *zyd* seizures. Together, these findings suggest glial $Ca^{2+}$ interfaces with calcineurin-dependent endocytosis to regulate plasma membrane protein levels and the $K^+$ buffering capacity of glia associated with neuronal somas. Disruption of these pathways leads to enhanced neuronal excitability and seizures, suggesting potential targets for future glial-based therapeutic modifiers of epilepsy.

## Results

### Mutations in a cortex glial NCKX generate stress-induced seizures without affecting brain structure or baseline neuronal function

We previously identified and characterized a *Drosophila* temperature-sensitive (TS) mutant termed *zydeco* (zyd) that exhibits seizure-like behavior (*Figure 1—video 1*, hereafter referred to as 'seizures', see Materials and methods for definition) when exposed to a variety of environmental stressors, including heat-shock and acute vortex. The *zyd* mutation disrupts a NCKX exchanger that extrudes cytosolic $Ca^{2+}$. Restoring zyd function specifically in cortex glial completely reverses the *zyd* seizure phenotype (*Melom and Littleton, 2013*). Cortex glia exhibit spatial segregation reminiscent of mammalian astrocytes, with each glial cell ensheathing multiple neuronal somas (*Awasaki et al., 2008*; *Melom and Littleton, 2013*). However, little is known about their role in the mature nervous system. In vivo $Ca^{2+}$ imaging using a myristoylated $Ca^{2+}$ sensitive-GFP (myrG-CaMP5) revealed small, rapid cortex glial $Ca^{2+}$ oscillations in wildtype *Drosophila* larvae (*Figure 1—video 2*, *Figure 1—figure supplement 1A*). In contrast, *zyd* mutants lack microdomain $Ca^{2+}$ transients and exhibit elevated baseline intracellular $Ca^{2+}$ (*Figure 1—video 3*), indicating altered glial $Ca^{2+}$ regulation underlies seizure susceptibility in *zyd* mutants (*Melom and Littleton, 2013*).

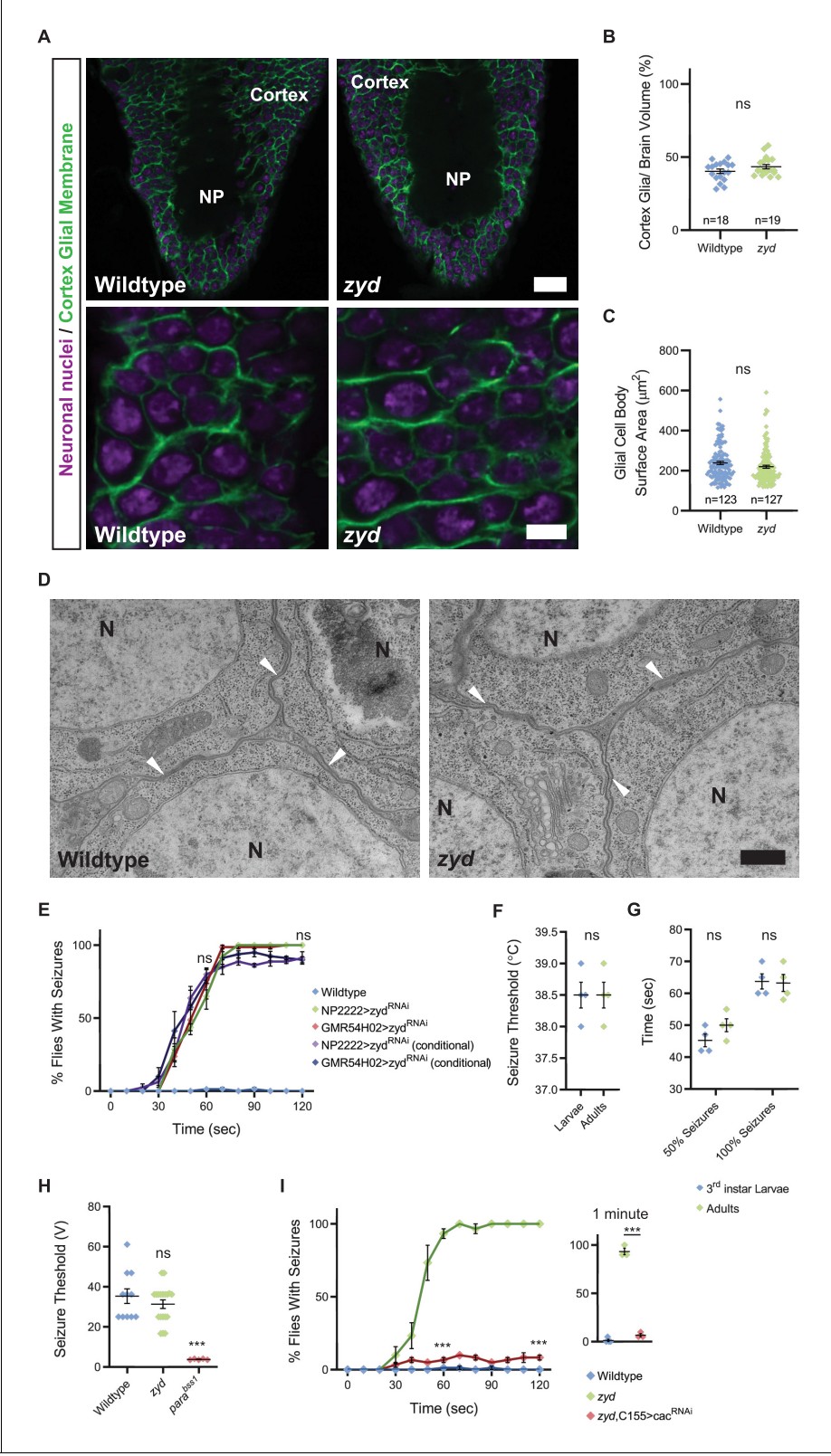

**Figure 1.** Mutations in a cortex glial NCKX generate stress-induced seizures. (**A**) Immunofluorescence imaging reveals no apparent morphological changes in cortex glial wrapping of neuronal soma (3rd instar larval brains, magenta: anti-Elav, neuronal nuclei; green: anti-GFP, mCD8:GFP, cortex glial membrane). Upper panels show a section through the VNC cortex and neuropil (NP), Scale bar = 20 μm. Lower panels show enlarged cortical regions. Scale bar = 5 μm. (**B**) Quantification of cortex volume occupied by cortex glial processes shows no difference between wildtype and *zyd* (n > 15 brains

*Figure 1 continued on next page*

*Figure 1 continued*

for each genotype, p=0.138). (C) Quantification of cortex glial cell body surface area shows no difference between wildtype and *zyd* (n > 120 cells/ N = 4 animals for each genotype, p=0.0892). (D) Electron microscopy images of cortex glial contacts (arrowheads) with neuronal somas (N). Cortex glial processes between neuronal cell bodies are as thin as 50 nm in both wildtype and *zyd*. Scale bar = 500 nm. (E) Time course of Heat-shock induced seizures (38.5°C, HS) following chronic or conditional knockdown of zyd with two different cortex glial drivers (NP2222 and GMR54H02) is shown. Rearing adult flies at the restrictive temperature (>30°C) for gal80$^{ts}$ (a temperature-sensitive form of the gal4 inhibitor, gal80, see Materials and methods) removes gal80 inhibition of gal4 and allows expression of zyd$^{RNAi}$ only at the adult stage. These manipulations reproduce the *zyd* mutant seizure phenotype (N = 4 groups of 20 flies/genotype). (F–G) Behavioral analysis of HS-induced seizures at 38.5°C shows that larval and adult seizures have similar temperature threshold (F) and kinetics (G) (N = 4 groups of 10–20 animals/condition/treatment). (H) Recordings of the giant fiber system muscle output. Seizure thresholds in wildtype, *zyd* and Para$^{bss1}$ (positive control) are shown. The voltage required to induce seizures in *zyd* is not significantly different from wildtype (35.32 ± 3.65V and 31.33 ± 2.12V, p=0.3191, n ≥ 7 flies/genotype). (I) Behavioral analysis of the time course of HS-induced seizures indicates neuronal knockdown of cac (C155>cac$^{RNAi}$) rescues the *zyd* seizure phenotype. Inset shows results after 1 minute of HS (p=0.0004, N = 4 groups of 20 flies/genotype). Error bars are SEM, ***=P < 0.001, Student's t-test.

DOI: https://doi.org/10.7554/eLife.44186.002

The following video and figure supplement are available for figure 1:

**Figure supplement 1.** Mutations in a cortex glial NCKX generate stress-induced seizures without affecting brain structure or baseline neuronal function.

DOI: https://doi.org/10.7554/eLife.44186.003

**Figure 1—video 1.** The response of wildtype flies to a 38.5°C heat-shock is shown, following by the response of *zyd* flies to the same condition.

DOI: https://doi.org/10.7554/eLife.44186.004

**Figure 1—video 2.** Representative Ca$^{2+}$ imaging in wildtype cortex glia.

DOI: https://doi.org/10.7554/eLife.44186.005

**Figure 1—video 3.** Representative Ca$^{2+}$ imaging in *zyd* cortex glia.

DOI: https://doi.org/10.7554/eLife.44186.006

Given cortex glia regulate the guidance of secondary axons and maintenance of cortical structural integrity (*Coutinho-Budd et al., 2017*; *Dumstrei et al., 2003*; *Spindler et al., 2009*), we first tested *zyd* larvae for morphological brain changes. Examination of brain structure and cortex glial morphology using fluorescent microscopy revealed no apparent changes in *zyd* mutants (*Figure 1A*), which showed similar cortex volume occupied by glial processes (*Figure 1B*) and cell body volume (*Figure 1C*) compared to wildtype cortex-glia. Closer examination of cortex glial process contacts with neuronal cell bodies using electron microscopy did not reveal morphological changes between controls and *zyd* mutants (*Figure 1D*). To determine if loss of ZYD affected glial or neuronal cell survival, we quantified the cell death marker DCP-1 (cleaved death caspase protein-1 [*Akagawa et al., 2015*]) in control and *zyd* 3$^{rd}$ instar larvae and adults. No change in cleaved DCP1 levels were found, indicating basal cell death was unaffected (*Figure 1—figure supplement 1B*). These data suggest mutations in *NCKX$^{zydeco}$* disrupt cortex glia function rather than morphology or development.

We previously found that conditionally restoring zyd function only during adult stages can reverse the *zyd* seizure phenotype (*Melom and Littleton, 2013*), providing additional evidence that the phenotype does not arise secondary to developmental changes or defective assembly of brain circuits. In a complementary approach, we knocked down zyd chronically throughout development or conditionally in adult stages following brain development using a UAS-zyd$^{RNAi}$ hairpin expressed with cortex glial-specific drivers (NP2222-gal4 and GMR54H02-gal4). ZYD was previously shown to be specifically required in cortex glia for the generation of seizures (*Melom and Littleton, 2013*). Both chronic and inducible cortex glial knockdowns mimicked the *zyd* TS seizure phenotype (*Figure 1E*). Seizure characteristics, including temperature threshold for seizure initiation and seizure kinetics, were similar between 3$^{rd}$ instar larvae and adults (*Figure 1F–G*), indicating a comparable requirement for zyd at both stages. Together, these results suggest that the *zyd* TS seizure phenotype is not due to morphological or developmental changes in brain anatomy, or changes in the ability of cortex glia to ensheath neuronal cell bodies.

NCKX transporters use the Na$^+$ and K$^+$ gradients to extrude Ca$^{2+}$, suggesting the loss of ZYD might alter the ionic balance of these ions that could affect neuronal membrane properties. Hence, we assayed if *zyd* animals displayed altered behaviors in the absence of the temperature trigger needed to induce seizures. We used a gentle touch assay (*Ma et al., 2016*; *Zhou et al., 2012*) to investigate whether the *zyd* mutation changed larval startle-induced behaviors, as elevated Ca$^{2+}$ activity in astrocytes was reported to correlate with elevated arousal in mice (*Ding et al., 2013*;

*Paukert et al., 2014*; *Srinivasan et al., 2015*) and in *Drosophila* (*Ma et al., 2016*). Crawling 3[rd] instar larvae touched anteriorly execute one of two responses: pausing and/or continuing forward (type I response) or an escape response by crawling backwards (type II response). We found that wildtype, *zyd* and NP2222>zyd[RNAi] larvae exhibited similar frequencies of type I and type II responses (*Figure 1—figure supplement 1C*). In addition, adult *zyd* flies exhibited normal locomotion (*Figure 1—figure supplement 1D*) and larvae exhibited normal light avoidance responses at room temperature (*Figure 1—figure supplement 1E*), indicating baseline neuronal properties required for these behaviors are unaffected. *Zyd* mutants also showed normal voltage thresholds in classical assays for giant fiber seizure induction, in contrast to animals harboring bang-sensitive mutations altering neuronal sodium channels (*Figure 1H*). Together with our previous observation that cortex glial knockdown of calmodulin (cam [*Melom and Littleton, 2013*]) completely reverses *zyd* seizures, these data indicate *zyd* mutants are unlikely to display baseline changes in ionic balance that alter intrinsic neuronal properties without the elevated temperature or vortex-induced hyperactivity.

To determine if elevated neuronal activity is required for the stress-induced seizures in *zyd* mutants, we assayed seizure behavior in animals with reduced synaptic transmission and neuronal activity. Pan-neuronal knockdown of Cacophony (cac), the presynaptic voltage-gated $Ca^{2+}$ channel responsible for neurotransmitter release (*Kawasaki et al., 2004*; *Rieckhof et al., 2003*), significantly reduced locomotion (*Figure 1—figure supplement 1F*) and rescued *zyd* TS-induced seizures (*Figure 1I*). These findings indicate elevated neuronal activity in *zyd* mutants during the heat shock is required for seizure induction following dysregulation of cortex glial $Ca^{2+}$.

## A genetic modifier screen of the *zyd* seizure phenotype reveals glia to neuron signaling mechanisms

To elucidate pathways by which cortex glial $Ca^{2+}$ signaling controls somatic regulation of neuronal function and seizure susceptibility, we performed a targeted RNAi screen for modifiers of the *zyd* TS seizure phenotype in adult animals. We reasoned that removal of a gene product required for this signaling pathway would prevent *zyd* TS seizures when absent. We used the pan-glial driver repo-gal4 to express RNAi to knockdown 847 genes encoding membrane receptors, secreted ligands, ion channels and transporters, vesicular trafficking proteins and known cellular $Ca^{2+}$ homeostasis and $Ca^{2+}$ signaling pathway components (*Supplementary files 1*, *2*). Given the broad role of $Ca^{2+}$ as a regulator of intracellular biology, we expected elevated $Ca^{2+}$ levels in *zyd* mutants to interface with several potential glial-neuronal signaling mechanisms. Indeed, the screen revealed multiple genetic interactions, identifying gene knockdowns that completely (28) or partially (21) rescued *zyd* seizures, caused lethality on their own (95) or synthetic lethality in the presence of the *zyd* mutation (5) or enhanced *zyd* seizures (37, *Supplementary file 1*).

Given TS seizures in *zyd* mutants can be fully rescued by reintroducing wildtype ZYD in cortex glia, we expected the genes identified in the RNAi pan-glial knockdown screen to function specifically within this population of cells. To assay cell-type specificity of the suppressor hits, we knocked down the top 33 rescue RNAis with cortex glial specific drivers (NP2222-gal4 and GMR54H02-gal4, *Supplementary file 1*). For the majority of suppressors, rescue with cortex glial specific drivers was weaker, either due to lower RNAi expression levels compared to the stronger repo-gal4 driver, or due to the requirement of the gene in other glial subtypes as well. To validate the rescue effects we observed, non-overlapping RNAis or mutant alleles for these genes were assayed (*Supplementary file 2*). For the current analysis, we focused only on the characterization of cortex glial $Ca^{2+}$-dependent pathways that are mis-regulated in *zyd*, and how this mis-regulation promotes neuronal seizure susceptibility.

## Cortex glial calcineurin activity is required for seizures in *zyd* mutants

We previously observed that knockdown of glial calmodulin (cam) eliminates the *zyd* seizure phenotype (*Melom and Littleton, 2013*), suggesting a $Ca^{2+}$/cam-dependent signaling pathway regulates glial to neuronal communication. Cam is an essential $Ca^{2+}$-binding protein that regulates multiple $Ca^{2+}$-dependent cellular processes and is abundantly expressed in *Drosophila* glia (*Altenhein et al., 2006*), although its role in glial biology is unknown. In the RNAi screen for *zyd* interactors, pan-glial knockdown of the regulatory calcineurin (CN) B subunit, CanB2, completely rescued both heat-shock

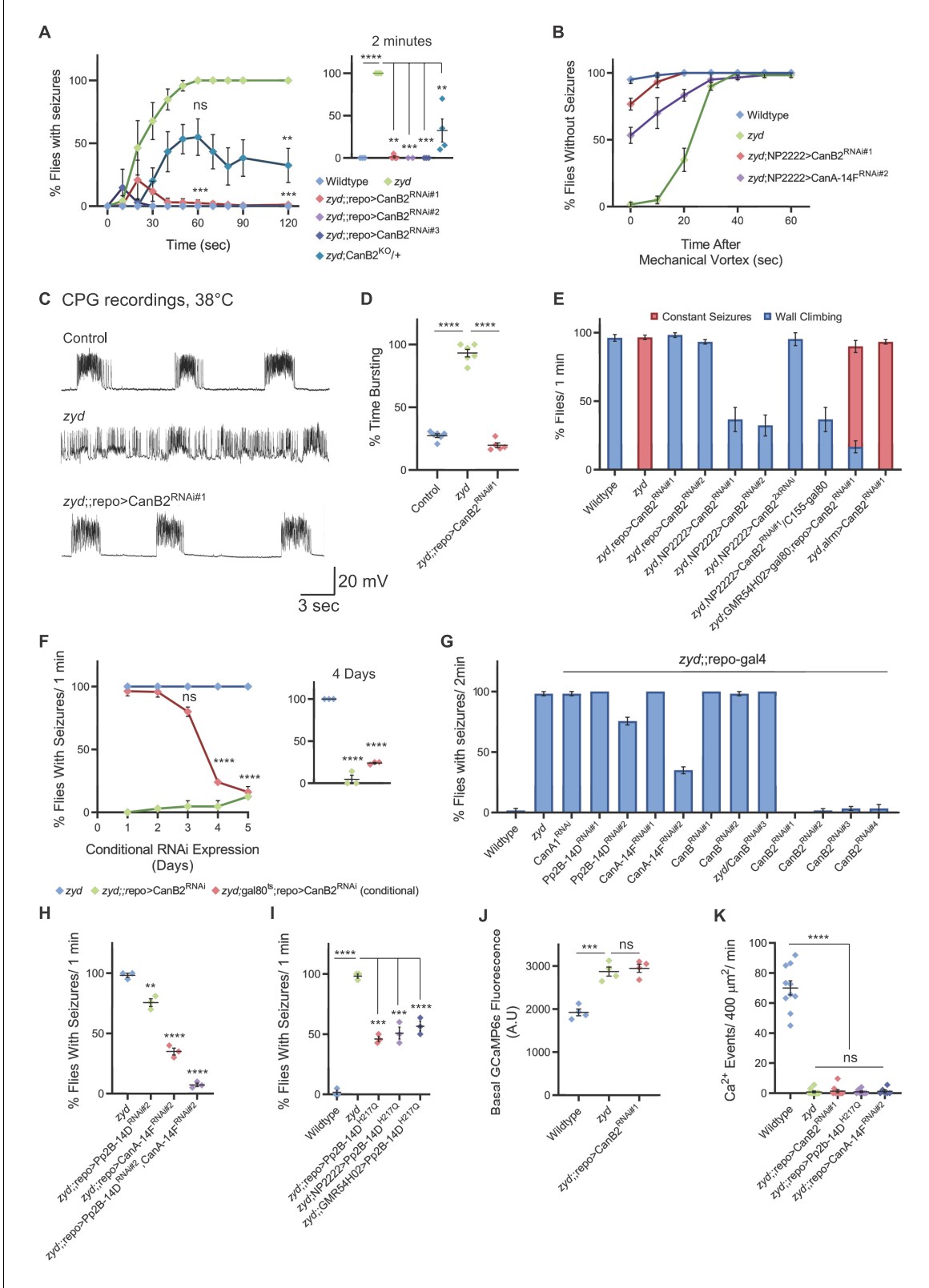

**Figure 2.** Cortex glial knockdown of calcineurin rescues *zyd* seizures without affecting intracellular Ca²⁺. (**A**) Behavioral analysis of HS-induced seizures. Pan-glial knockdown of the CN regulatory subunit, CanB2, with three partially overlapping hairpins (#1, #2 and #3, see Materials and methods) completely rescues the *zyd* seizure phenotype, while a single copy of CanB2 knockout allele (CanB2$^{KO}$/+) rescues ~60% of seizures (N = 4 groups of ≥15 flies/genotype). Inset shows analysis after 2 minutes of HS (p=0.0001). (**B**) Behavioral analysis of the recovery from vortex-induced seizures. Pan-

*Figure 2 continued on next page*

*Figure 2 continued*

glial knockdown of CanB2 and CanA-14F rescues *zyd* vortex-induced seizures (N = 3 groups of 20 flies/genotype). (C) Representative voltage traces of spontaneous CPG activity at larval 3$^{rd}$ instar muscle 6 at 38°C in wildtype, *zyd* and *zyd*;;repo >CanB2$^{RNAi\#1}$ animals ($n \geq 5$ preparations/genotype). (D) Quantification of average bursting duration for CPG recordings of the indicated genotypes at 38°C ($n \geq 5$ preparations/genotype). (E) Detailed analysis (see Materials and methods) of HS induced behaviors of *zyd*/CanB2$^{RNAi}$ flies. Cortex glial knockdown of CanB2 leads to seizure rescue in ~30% of *zyd*; NP2222>CanB2$^{RNAi}$ flies, with the remaining ~70% displaying partial rescue. Cortex glial CanB2 knockdown with two copies of the RNAi (*zyd*; NP2222>CanB2$^{2xRNAi}$) recapitulates the full rescue seen with pan-glial knockdown. Inhibiting gal4 expression of the RNAi in neurons with gal80 (C155-gal80) does not alter the rescue observed with cortex glial knockdown, and astrocyte specific (alrm-gal4) CanB2 knockdown does not rescue *zyd* seizures (N = 3 groups of >15 flies/genotype, see *Figure 2—figure supplement 1D* for complete dataset). (F) Cortex glial conditional knockdown of CanB2 using gal4/gal80$^{ts}$. Rearing adult flies at the restrictive temperature (>30°C) for gal80$^{ts}$ allows expression of CanB2$^{RNAi}$ only at the adult stage. A significant reduction in seizures (p<0.0001) was seen after four days of rearing flies at the restrictive temperature for gal80$^{ts}$ (31°C), with only ~25% of adults showing seizures. The reduction in seizures was enhanced when adults were incubated at 31°C for longer periods (N = 3 groups of >10 flies/ genotype). Inset shows analysis after 4 days of incubation at 31°C (p=0.0001). (G) Pan-glial knockdown of the *Drosophila* calcineurin (CN) family (CanA1, CanA-14D/Pp2B-14D, CanA-14F, CanB and CanB2) indicate CanB2 knockdown completely rescues *zyd* seizures, CanA-14D and CanA-14F knockdowns partially reduce seizures (N = 4 groups of >10 flies/genotype). (H) Pan-glial knockdown of Pp2B-14D and CanA-14F partially rescues the *zyd* HS seizures phenotype (~25% rescue for Pp2B-14D, p=0.0032; and ~60% rescue for CanA14F, p<0.0001). Knocking down the two genes simultaneously rescues *zyd* seizures, with only ~10% of flies showing seizures (~90% rescue, p<0.0001, N = 3 groups of >10 flies/genotype). (I) Overexpressing a dominant-negative form on Pp2B-14D (CanA$^{H217Q}$) rescues ~50% of *zyd* seizures regardless of the driver used (repo: p<0.0001; NP2222: p=0.0006; GMR54H02- p=0.0004. N = 3 groups of >10 flies/genotype). (J) Larval Ca$^{2+}$ imaging in cortex glia expressing myrGCaMP6s indicates the elevated basal Ca$^{2+}$ fluorescence at 25°C observed in *zyd* mutants relative to wildtype cortex glia (p=0.0003) is not altered following CanB2 knockdown (*zyd*;;repo>CanB2$^{RNAi}$, p=0.6096. $n \geq 5$ animals/genotype). (K) Microdomain Ca$^{2+}$ oscillations observed in wildtype cortex glia expressing myrGCaMP6s are abolished in *zyd* cortex glia and are not restored following either CanB2 or CanA14F knockdown ($n \geq 5$ animals/genotype). Error bars are SEM, **=P < 0.01, ***=P < 0.001, ****=P < 0.0001, Student's t-test.

DOI: https://doi.org/10.7554/eLife.44186.007

The following video and figure supplement are available for figure 2:

**Figure supplement 1.** Cortex glial knockdown of calcineurin rescues *zyd* seizures without affecting intracellular Ca$^{2+}$.

DOI: https://doi.org/10.7554/eLife.44186.008

**Figure 2—video 1.** The response of zyd/repo>CanB2$^{RNAi}$ flies to a 38.5°C heat-shock is shown.

DOI: https://doi.org/10.7554/eLife.44186.009

**Figure 2—video 2.** The response of zyd/NP2222>CanB2$^{RNAi}$ flies to a 38.5°C heat-shock is shown.

DOI: https://doi.org/10.7554/eLife.44186.010

**Figure 2—video 3.** The response of zyd/NP2222>CanB2$^{2xRNAi}$ flies to a 38.5°C heat-shock is shown.

DOI: https://doi.org/10.7554/eLife.44186.011

**Figure 2—video 4.** Representative Ca$^{2+}$ imaging in repo>CanB2$^{RNAi}$ cortex glia.

DOI: https://doi.org/10.7554/eLife.44186.012

**Figure 2—video 5.** Representative Ca$^{2+}$ imaging in zyd/repo>CanB2$^{RNAi}$ cortex glia.

DOI: https://doi.org/10.7554/eLife.44186.013

and vortex induced seizures in *zyd* animals (*Figure 2A–B*, *Figure 2—video 1*). Recordings of the motor central pattern generator (CPG) muscle output at the larval neuromuscular junction (NMJ), showed that in contrast to the continuous neuronal firing observed in *zyd* mutants, recordings from *zyd*;;repo >CanB2$^{RNAi\#1}$ larvae exhibit normal rhythmic firing at 38°C similar to wildtype controls (*Figure 2C,D*). The rescue effect of CanB2 knockdown on *zyd* seizure phenotype was similar when CanB2 was targeted using three additional, partially-overlapping CanB2 RNAi constructs (*Figure 2A,G*). In addition, the *zyd* seizure phenotype was partially rescued when combined with a heterozygous CanB2 knockout (*zyd*;CanB2$^{KO}$/+) allele (*Nakai et al., 2011*; *Figure 2A*, CanB2$^{KO}$ homozygotes are lethal and could not be tested). Knocking down CanB2 on a wildtype background was viable (*Figure 2—figure supplement 1A*) and did not cause any significant change in larval light avoidance (*Figure 2—figure supplement 1B*) or adult locomotion and activity (*Figure 2—figure supplement 1C*). To refine the glial subpopulation in which CanB2 activity is necessary to promote seizures in *zyd* mutants, we knocked down CanB2 using glial subtype specific drivers. CanB2 knockdown in astrocytes resulted in no rescue of the *zyd* phenotype, while CanB2 knockdown with a cortex-glial specific driver (NP2222-gal4) greatly improved the *zyd* phenotype (*Figure 2E*, *Figure 2—figure supplement 1D*). Animals no longer displayed continuous seizures, although the rescue was less robust compared to pan-glial knockdown (*Figure 2E*, *Figure 2—figure supplement 1D*, *Figure 2—video 2*), possibly due to lower expression level of the RNAi. Indeed, rescue was greatly enhanced by cortex glial-specific knockdown of CanB2 using two copies of the CanB2 RNAi

construct (*Figure 2E*, *Figure 2—video 3*), with ~90% of *zyd*;NP2222>CanB2[2xRNAi] adult animals lacking seizures. The effect of CanB2 knockdown was specific to cortex glia, as it was insensitive to blockade of expression of CanB2[RNAi] in neurons using C155–gal80 (a neuron specific gal4 repressor; *Figure 2E* and *Figure 2—figure supplement 1D*). To exclude a developmental effect of CanB2[RNAi] knockdown within glia, we conditionally expressed a single copy of CanB2[RNAi] (with gal4/gal80[ts], see Materials and methods) only in adult *zyd* mutant flies. Adult flies reared at the permissive temperature for gal80[ts] (>30 °C) to allow CanB2[RNAi] expression exhibited significantly fewer seizures after 3 days, with only ~20% of flies displaying *zyd*-like seizures by 5 days (*Figure 2F*). *Zyd* seizure rescue by CanB2 knockdown did not result from simple alterations in motility, as repo>CanB2[RNAi] and *zyd*;;repo>CanB2[RNAi] animal exhibited normal larval light avoidance response (*Figure 2—figure supplement 1E*) and adult locomotion (*Figure 2—figure supplement 1F*). We conclude that CanB2 is required in cortex glia to promote *zyd* TS seizure activity.

Calcineurin (CN) is a highly conserved $Ca^{2+}$/cam-dependent protein phosphatase implicated in a number of cellular processes (*Rusnak and Mertz, 2000*). CN is a heterodimer composed of a ~60 kDa catalytic subunit (CanA) and a ~19 kDa EF-hand $Ca^{2+}$-binding regulatory subunit (CanB). Both subunits are essential for CN phosphatase activity. The *Drosophila* CN gene family contains three genes encoding CanA (CanA1, Pp2B-14D and CanA-14F) and two genes encoding CanB (CanB and CanB2) (*Takeo et al., 2006*). Previous studies in *Drosophila* demonstrated several CN subunits (CanA-14F, CanB, and CanB2) are broadly expressed in the adult *Drosophila* brain (*Tomita et al., 2011*) and that neuronal CN is essential for regulating sleep (*Nakai et al., 2011*; *Tomita et al., 2011*). CN function within glia has not been characterized. We found that pan-glial knockdown of two CanA subunits, Pp2B-14D and CanA-14F, partially rescued *zyd* heat-shock and vortex-induced seizures (*Figure 2B,G*). The rescue was more robust for vortex-induced seizures than those induced by heat-shock, suggesting heat-shock is likely to be a more severe hyperexcitability trigger (*Figure 2B*). Rescue was enhanced by knockdown of both Pp2B-14D and CanA-14F (*Figure 2H*), with more than ~90% of *zyd*;;repo>Pp2B-14D[RNAi],CanA-14F[RNAi] flies lacking seizures, suggesting a redundant function of these two subunits in glial cells. Similar to CanB2, knockdown of both Pp2b-14D and CanA-14F on a wildtype background was viable (*Figure 2—figure supplement 1A*) and did not cause any significant change in larval light avoidance (*Figure 2—figure supplement 1B*) or adult locomotion and activity (*Figure 2—figure supplement 1C*). We next overexpressed a dominant negative form of CanA (Pp2B-14D[H217Q]) using either pan-glial (repo-gal4) or two different cortex-glial specific drivers (NP2222-gal4 and GMR54H02-gal4). Overexpressing Pp2B-14D[H217Q] resulted in ~50% of *zyd*/Pp2B-14D[H217Q] flies becoming seizure-resistant regardless of the driver used (*Figure 2I*). These results indicate CN activity is required in cortex glia to promote seizures in *zyd* mutants. Imaging intracellular $Ca^{2+}$ in cortex glia with GMR54H02-gal4 driving myrGCaMP6s revealed that CN knockdown had no effect on wildtype cortex glial $Ca^{2+}$ oscillatory behavior or the elevated basal $Ca^{2+}$ levels and the lack of microdomain $Ca^{2+}$ events in *zyd* cortex glia (*Figure 2J–K*, *Figure 2—video 4* and *5*). These observations indicate CN function is required downstream of elevated intracellular $Ca^{2+}$, rather than to regulate $Ca^{2+}$ influx or efflux in cortex glial cells. Together, these results demonstrate that a CN-dependent signaling mechanism in cortex glia is required for a glia to neuronal pathway that drives seizure generation in *zyd* mutants.

## Calcineurin activity is enhanced in *zyd* cortex glia

To characterize CN activity in wildtype and *zyd* cortex glia we used the CalexA ($Ca^{2+}$-dependent nuclear import of LexA) system (*Masuyama et al., 2012*) as a reporter for CN activity. In this assay, sustained neural activity induces CN activation and dephosphorylation of a chimeric transcription factor LexA-VP16-NFAT (termed CalexA) which is then transported into the nucleus. The imported dephosphorylated CalexA drives GFP reporter expression (*Figure 3—figure supplement 1*). The CalexA components were brought into control and *zyd* mutant backgrounds to directly assay CN activity. A substantial basal activation of CN was observed in control 3[rd] instar larval cortex glia at room temperature using fluorescent imaging (*Figure 3A*). CN activity and the resulting GFP expression was enhanced in *zyd* cortex glia (*Figure 3B*) and greatly reduced in *zyd*/CanB2[RNAi] cortex glia (*Figure 3C–C'*). Western blot analysis of CalexA-induced GFP expression in adult head extracts revealed enhanced cortex glial CN activity in adult *zyd* mutants compared to controls (23 ± 3% enhancement, *Figure 3D–E*). RNAi knockdown of CanB2 reduced CalexA GFP expression as expected (27 ± 1%, *Figure 3C–E*). These results demonstrate CN activity is enhanced downstream

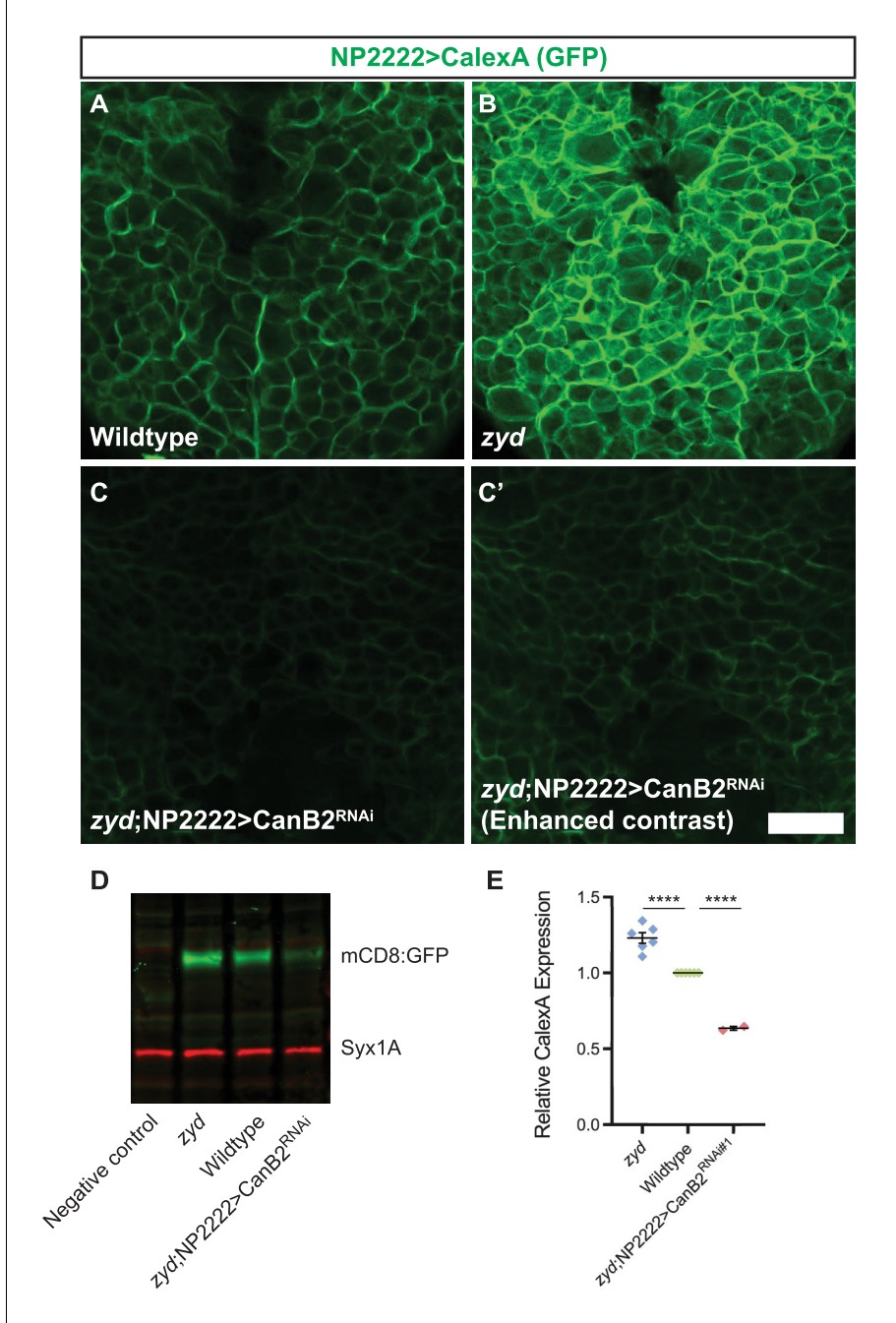

**Figure 3.** Calcineurin activity is enhanced in *zyd* cortex glia and can be efficiently suppressed by CanB2 knockdown. (**A-C**) Fluorescence microscopy imaging of cortex glial CalexA-derived GFP expression in wildtype (**A**), *zyd* (**B**) and *zyd*;NP2222>CanB2^RNAi (**C-C'**) larvae. Green: anti-GFP = cortex glial CN activity (animals were reared at 25°C, Scale bar = 20 μm, N ≥ 5 animals/genotype). (**D-E**) Western blot analysis of cortex glial CalexA derived GFP expression (NP2222>CalexA) in *zyd*, wildtype and *zyd*;NP2222>CanB2^RNAi adult heads. CN activity is enhanced by ~25% (p<0.0001) in *zyd* cortex glia and reduced by ~35% (p<0.0001) in CanB2^RNAi animals (N ≥ 2 experiment, five heads/sample). GFP signals in each experiment were normalized to wildtype. Error bars are SEM, ****=P < 0.0001, Student's t-test.

DOI: https://doi.org/10.7554/eLife.44186.014

The following figure supplement is available for figure 3:

**Figure supplement 1.** Schematic representation of the CalexA system.

DOI: https://doi.org/10.7554/eLife.44186.015

of the elevated Ca$^{2+}$ levels in *zyd* mutant cortex glia, and that CN activity can be efficiently reduced by RNAi knockdown of CanB2.

## Pharmacological inhibition of calcineurin rescues *zyd* seizures

Several seizure mutants in *Drosophila* can be suppressed by commonly used anti-epileptic drugs (*Kuebler and Tanouye, 2002*; *Song and Tanouye, 2008*), indicating conservation of key mechanisms that regulate neuronal excitability. The catalytic activity of CanA is strictly controlled by Ca$^{2+}$ levels, calmodulin, and CanB, and can be inhibited by the immunosuppressants cyclosporine A (CsA) and FK506. To assay if *zyd* TS seizures can be prevented with an anti-CN drug, adult flies were fed with media containing CsA and tested for HS induced seizures after 0, 3, 6, 12 and 24 hr of drug feeding (red arrowheads in *Figure 4A*, *Figure 4B–D*). *Zyd* flies fed with 1 mM CsA for 12 hr showed ~80% fewer seizures than controls (*Figure 4B–D*). Seizure rescue by CsA was dose-

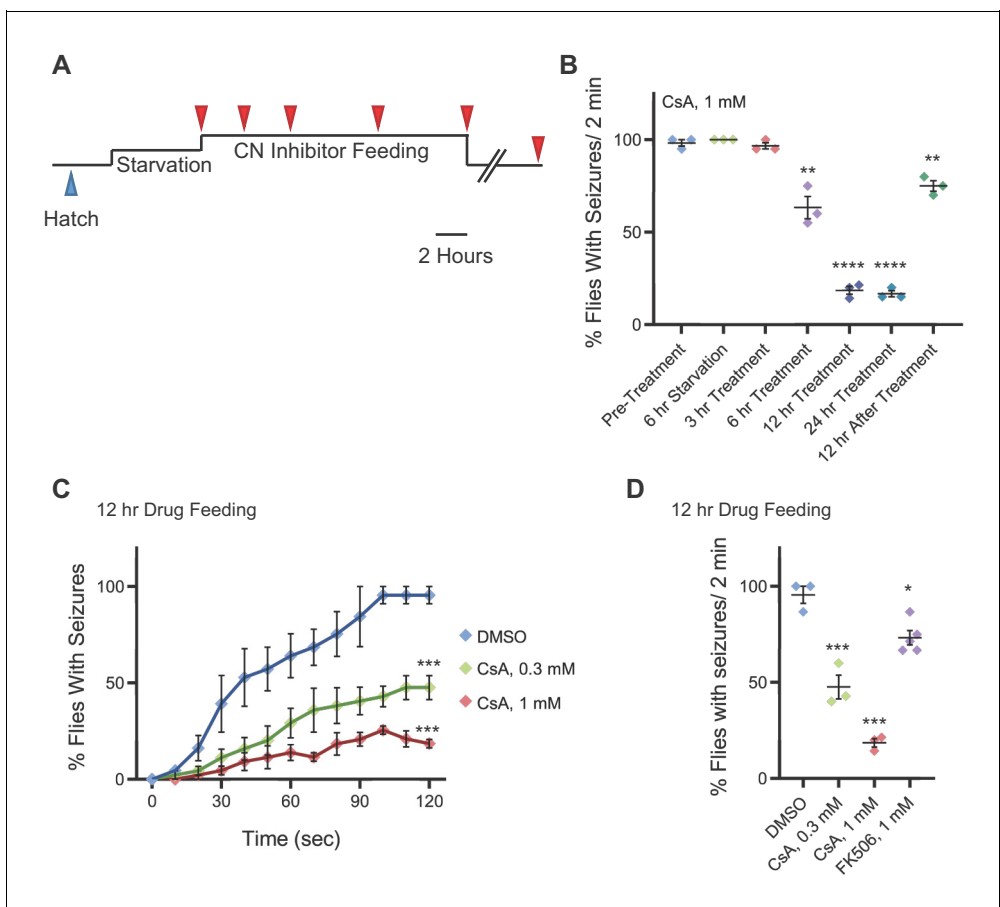

**Figure 4.** Pharmacologically targeting calcineurin activity suppresses *zyd* heat shock-induced seizures. (**A**) Schematic representation of the experimental design. Adult male flies (<1 day old) were starved for 6 hours and fed with liquid medium containing CN inhibitors for 3, 6, 12 or 24 hours (red arrowheads), before testing for HS induced seizures. Flies were also tested 12 and 24 hours after drug withdrawal. (**B-D**) Behavioral analysis of HS induced seizures. (**B**) Summary of all time points for CsA treatment (N = 3 groups of 15–20 flies/treatment. 6 hours feeding: p=0.005; 12/24 hours feeding: p<0.0001; 12 hours drug withdrawal: p=0.0022). (**C**) Flies were fed with 0.3 mM or 1 mM of CsA for 12 hours. Feeding with 1 mM CsA reduces seizures by ~75% (p<0.0001). The effect of CsA treatment on HS-induced seizures shows a significant dose-dependent reduction in seizure occurrence (N = 3 groups of >15 flies/treatment). (**D**) After 2 minutes of heat-shock, seizures were reduced by ~50% (p=0.041) in flies that were fed with 0.3 mM CsA, and by ~80% (p=0.0062) in flies that were fed with 1 mM CsA. A ~ 20% (p=0.043) reduction in seizures was observed when flies were fed with 1 mM FK506 (N = 3 groups of >15 flies/treatment). Error bars are SEM, *=P < 0.05, **=P < 0.01, ****=P < 0.0001, Student's t-test.
DOI: https://doi.org/10.7554/eLife.44186.016

dependent, with less robust suppression when flies were fed with 0.3 mM CsA (*Figure 4B–D*). The CsA rescue was reversible, as seizures reoccurred following 12 hr of CsA withdrawal (*Figure 4B*). Feeding flies with a second anti-CN drug, FK506, resulted in a partial rescue of the phenotype (*Figure 4D*). Although we cannot exclude off-target effects of these compounds in *Drosophila*, these data suggest pharmacologically targeting the CN pathway can improve the outcome of glial-derived neuronal seizures in the *zyd* mutant.

## Cortex glial knockdown of the two-pore-domain K$^+$ channel, sandman, mimics *zyd* seizures

To explore how CN hyperactivation promotes seizures, we conducted a screen of known and putative CN targets using RNAi knockdown with repo-gal4. We concentrated our screen on putative CN target genes that are involved in signal transduction (*Supplementary file 3*). This screen revealed that pan-glial knockdown of sandman (sand), the *Drosophila* homolog of TRESK (KCNK18) and a member of the two-pore-domain K$^+$ channel family (K$_{2P}$), caused adult flies to undergo TS-induced seizures similar to *zyd* mutants (*Figure 5A*, *Figure 5—video 1*). Vortex-induced seizures in repo>sand$^{RNAi}$ were less severe than those observed in *zyd*, with only ~50% of sand$^{RNAi}$ flies showing seizures (*Figure 5—figure supplement 1A*). TS-induced seizures in repo>sand$^{RNAi}$ adults were found to have the same kinetics and temperature threshold as seizures observed in *zyd* mutants (*Figure 5A, B*), and CPG recordings showed that repo>sand$^{RNAi}$ larvae exhibit rapid, unpatterned firing at 38°C, similar to *zyd* (*Figure 5C,D*). Cortex-glial specific knockdown of sand recapitulated ~50% of the seizure effect when two copies of the RNAi were expressed (*Figure 5A,B*). The less robust effect observed with the cortex-glial driver could be due to less effective RNAi knockdown or secondary to a role for sand in other glial subtypes. To determine if sand functions in other glia subtypes to mimic the *zyd* seizure pathway, we expressed sand$^{RNAi}$ using the pan-glial driver repo-gal4 and inhibited expression specifically in cortex glia with GMR54H02>gal80. In the absence of cortex glial-knockdown of sand, seizure generation was suppressed (*Figure 5A*). Similar to *zyd* mutants, sand$^{RNAi}$ animals did not show changes in general activity and locomotion at room temperature (*Figure 5E*). To assess whether the seizure phenotypes in *zyd* and sand$^{RNAi}$ originate from the same pathway, we tested flies in which the *zyd* mutation is combined with sand$^{RNAi}$. We tested different genetic combinations for seizure temperature threshold (*Figure 5B*), light avoidance (*Figure 5E*) and seizures kinetics (*Figure 5—figure supplement 1B*). Pan-glial or cortex glial knock down of sand did not enhance the hemizygote *zyd* phenotype, judged by seizure temperature threshold (*Figure 5B*) and kinetics (*Figure 5—figure supplement 1B*). Heterozygous *zyd* flies expressing two copies of the sand$^{RNAi}$ in cortex glia (zyd/+;NP2222>sand$^{2xRNAi}$) showed the same seizure frequency (~60% of flies seizing after two minutes of heat-shock), with similar seizures characteristics as those observed when only sand$^{2xRNAi}$ was expressed (*Figure 5B*, *Figure 5—figure supplement 1B*). Together, these results suggest seizures due to loss of sand and *zyd* impinge on a similar pathway.

Mammalian astrocytes modulate neuronal network activity through regulation of K$^+$ buffering (*Bellot-Saez et al., 2017*), in addition to their role in uptake of neurotransmitters such as GABA and glutamate (*Murphy-Royal et al., 2017*). Human K$_{ir4.1}$ potassium channels (*KCNJ10*) have been implicated in maintaining K$^+$ homeostasis, with mutations in the loci causing epilepsy (*Haj-Yasein et al., 2011*). In addition, gain and loss of astrocytic K$_{ir4.1}$ influence the burst firing rate of neurons through astrocyte-to-neuronal cell body contacts (*Cui et al., 2018*). However, K$_{ir}$ channels are unlikely to be the only mechanism for glial K$^+$ clearance, as K$_{ir4.1}$ channels account for less than half of the K$^+$ buffering capacity of mature hippocampal astrocytes (*Ma et al., 2014*). To determine if cortex glial K$_{ir}$ channels regulate seizure susceptibility in addition to sand, we used repo-gal4 to knock down all three *Drosophila* K$_{ir}$ family members (Irk1, Irk2 and Irk3). Pan-glial knockdown of the *Drosophila* K$_{ir}$ family did not cause seizures (*Figure 5—figure supplement 1C*), while knock down of either Irk1 or Irk2 slightly enhanced the *zyd* phenotype (*Figure 5—figure supplement 1D*). Similarly, repo-gal4 knockdown of other well-known *Drosophila* K$^+$ channels beyond the K$_{ir}$ family also did not cause seizures (*Figure 5—figure supplement 1C*), indicating sand is likely to play a preferential role in K$^+$ buffering in *Drosophila* glia.

The mammalian sand homolog, TRESK, is directly activated by CN dephosphorylation (*Czirják et al., 2004*; *Enyedi and Czirják, 2015*), while *Drosophila* sand was shown to be modulated in sleep neurons by activity-induced internalization from the plasma membrane (*Pimentel et al., 2016*). Regardless of the mechanism by which CN may regulate the protein, we hypothesized that

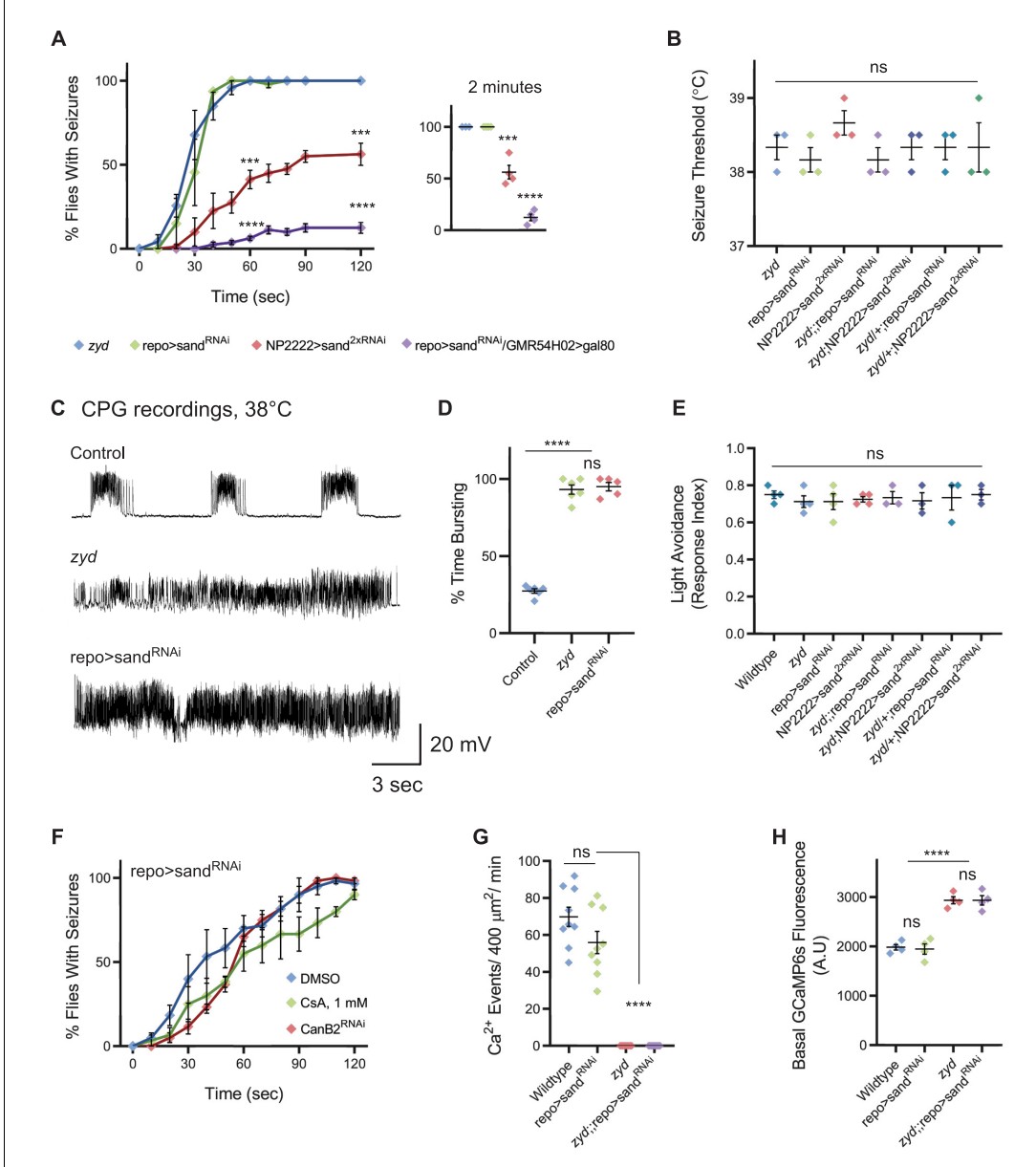

**Figure 5.** Cortex glial knock-down of sandman, a K$_{2P}$ channel, recapitulates *zyd* phenotypes. (**A-B**) Behavioral analysis of HS induced seizures. (**A**) Knockdown of sandman (sand) in different glial subtypes: pan-glial (repo), cortex glial (NP2222) and in all glia other than cortex glia (repo>sand$^{RNAi}$/GMR54H02>gal80 in which gal80 is constitutively inhibiting gal4 activity and sand$^{RNAi}$expression only in cortex glia). Inset shows analysis after 2 minutes of HS (p=0.0006 for NP2222>sand$^{2xRNAi}$, p<0.0001 for repo>sand$^{RNAi}$/GMR54H02>gal80, N = 4 groups of >10 flies/genotype). (**B**) Temperature threshold of repo>sand$^{RNAi}$ (p=0.5185) and NP2222>sand$^{2xRNAi}$ (p=0.2302) seizures in comparison to *zyd* (N = 3 groups of 10/temperature/genotype). (**C**) Representative voltage traces of spontaneous CPG activity at larval 3$^{rd}$ instar muscle 6 at 38°C in wildtype, *zyd* and repo>sand$^{RNAi}$ (n ≥ 5 preparations/genotype). (**D**) Quantification of average bursting duration for CPG recordings of the indicated genotypes at 38°C. (**E**) Light avoidance assay reveals no defect in this behavior at 25°C (N = 3 groups of 20 flies/genotype). (**F**) Behavioral analysis of HS-induced seizures. Seizures in repo>sand$^{RNAi}$ animals were not suppressed with either CanB2$^{RNAi#1}$ or by feeding flies with 1 mM CsA (N = 3 groups of 20 flies/genotype/treatment). (**G-H**) Ca$^{2+}$ imaging in larval cortex glial cells using myrGCaMP6s. (**G**) The average rate of microdomain Ca$^{2+}$ events was reduced in repo>sand$^{RNAi}$ cortex glia relative to wildtype (20.36 ± 5.5 and 69.83 ± 5.3, p<0.0001). Knockdown of sand on the *zyd* background did not restore *zyd* Ca$^{2+}$ microdomain events (n ≥ 5 animals/genotype). (**H**) Average myrGCaMP6s fluorescence in cortex glia at 25°C. Elevated basal fluorescence of GCaMP6s in *zyd* relative to wildtype cortex glia (p=0.0003) is not altered following sand knockdown (*zyd*;;repo>sand$^{RNAi}$, N = 4 animals/genotype. Error bars are SEM, ***=P < 0.001, ****=P < 0.0001, Student's t-test.

DOI: https://doi.org/10.7554/eLife.44186.017

The following video and figure supplement are available for figure 5:

*Figure 5 continued on next page*

*Figure 5 continued*

**Figure supplement 1.** Cortex glial knockdown of sandman, a $K_{2P}$ channel, reproduces *zyd* phenotypes.
DOI: https://doi.org/10.7554/eLife.44186.018

**Figure 5—video 1.** The response of repo>sand$^{RNAi}$ flies to a 38.5°C heat-shock is shown, following by the response of NP2222>sand$^{2xRNAi}$ flies to the same condition.
DOI: https://doi.org/10.7554/eLife.44186.019

**Figure 5—video 2.** Representative $Ca^{2+}$ imaging in repo>sand$^{RNAi}$ cortex glia.
DOI: https://doi.org/10.7554/eLife.44186.020

sand is epistatic to CN in controlling *zyd*-mediated seizures. Indeed, inhibition of CN by RNAi or CsA did not alter sand$^{RNAi}$-induced seizures (*Figure 5F*), placing sand downstream of CN activity. Furthermore, knockdown of sand in the *zyd* background does not alter the elevated basal $Ca^{2+}$ or the lack of microdomain $Ca^{2+}$ events in *zyd* mutants (*Figure 5G,H*), and cortex glial $Ca^{2+}$ oscillatory behavior in repo >sand$^{RNAi}$ is similar to wildtype (*Figure 5G,H* and *Figure 5—video 2*), suggesting sand is downstream to the abnormal $Ca^{2+}$ signaling in *zyd.* Overall, these findings suggest elevated $Ca^{2+}$ in *zyd* mutants leads to hyperactivation of CN and subsequent reduction in sand function. These results suggest that impairment in glial buffering of the rising extracellular $K^+$ during elevated neuronal activity and stress conditions (heat shock or acute vortex) causes enhanced seizure susceptibility in *zyd* mutants.

## SAND is not differentially phosphorylated in *zyd* cortex glia

We next sought to examine how elevated CN activity in *zyd* mutants alters sand function. We generated a transgenic *Drosophila* strain to express GFP-tagged sand specifically in cortex glia (UAS-sand:eGFP, see Materials and methods). First, we used these lines to verify sand knockdown by sand$^{RNAi}$, and found that SAND:GFP expression is reduced by ~45% when both sand and sand$^{RNAi}$ are expressed using a pan glial driver (repo-gal4), while the expression of a control RNAi did not change SAND:GFP expression (*Figure 6A,B*).

The mammalian sand homolog, TRESK, is constitutively phosphorylated on four serine residues (S264 by PKA, and S274, S276 and S279 by MARK1 [*Enyedi and Czirják, 2015*]). Two of these residues are conserved in *Drosophila* sand (S264 and S276, see *Figure 6—figure supplement 1A* for protein alignment). Constitutive dephosphorylation and subsequent activation of sand by CN is predicted to increase $K^+$ buffering following hyperactivation of the nervous system by stressors, and thus fewer seizures would be expected – opposite to what we have observed. To directly asses sand phosphorylation status in wildtype and *zyd* cortex glia, we used $Mn^{2+}$ phosphate binding tag (Phos-tag) gel electrophoresis (*Kinoshita et al., 2006*) to separate phosphorylated species of cortex glial SAND:GFP (expressed by the GMR54H02 driver). We found that SAND:GFP was not differentially phosphorylated in wildtype versus *zyd* mutants (*Figure 6C*). Multiple bands were detected when samples were pre-treated with alkaline phosphatase (*Figure 6C*, arrowheads), indicating SAND:GFP is indeed phosphorylated on several phosphorylation sites in cortex glia. This analysis also revealed that SAND:GFP expression level was reduced in *zyd* relative to wildtype cortex glia (see below). Together with the prediction that regulation of sand by dephosphorylation should lead to seizure suppression, these results argue against enhanced sand dephosphorylation as the primary cause of *zyd* seizures.

## Enhanced endocytosis in *zyd* cortex glia leads to reduction of plasma membrane SAND

A second mechanism to link elevated $Ca^{2+}$ and CN hyperactivation to sand regulation is suggested by a previous study demonstrating sand expression on the plasma membrane of neurons involved in sleep homeostasis in *Drosophila* is regulated by activity-dependent internalization (*Pimentel et al., 2016*). Cam and CN activate several endocytic $Ca^{2+}$ sensors and effectors that control $Ca^{2+}$-dependent endocytosis (*Xie et al., 2017*). If hyperactivity of CN leads to enhanced internalization of sand and subsequent seizure susceptibility due to decreased $K^+$ buffering capacity, we hypothesized that sand would be differentially distributed in *zyd* cortex glia. Indeed, examination of SAND:GFP distribution within cortex glial cells showed a significant reduction in SAND:GFP fluorescence on cortex glial membranes (53 ± 6.85% reduction, *Figure 6D,E*). Consistent with this observation, SAND:GFP

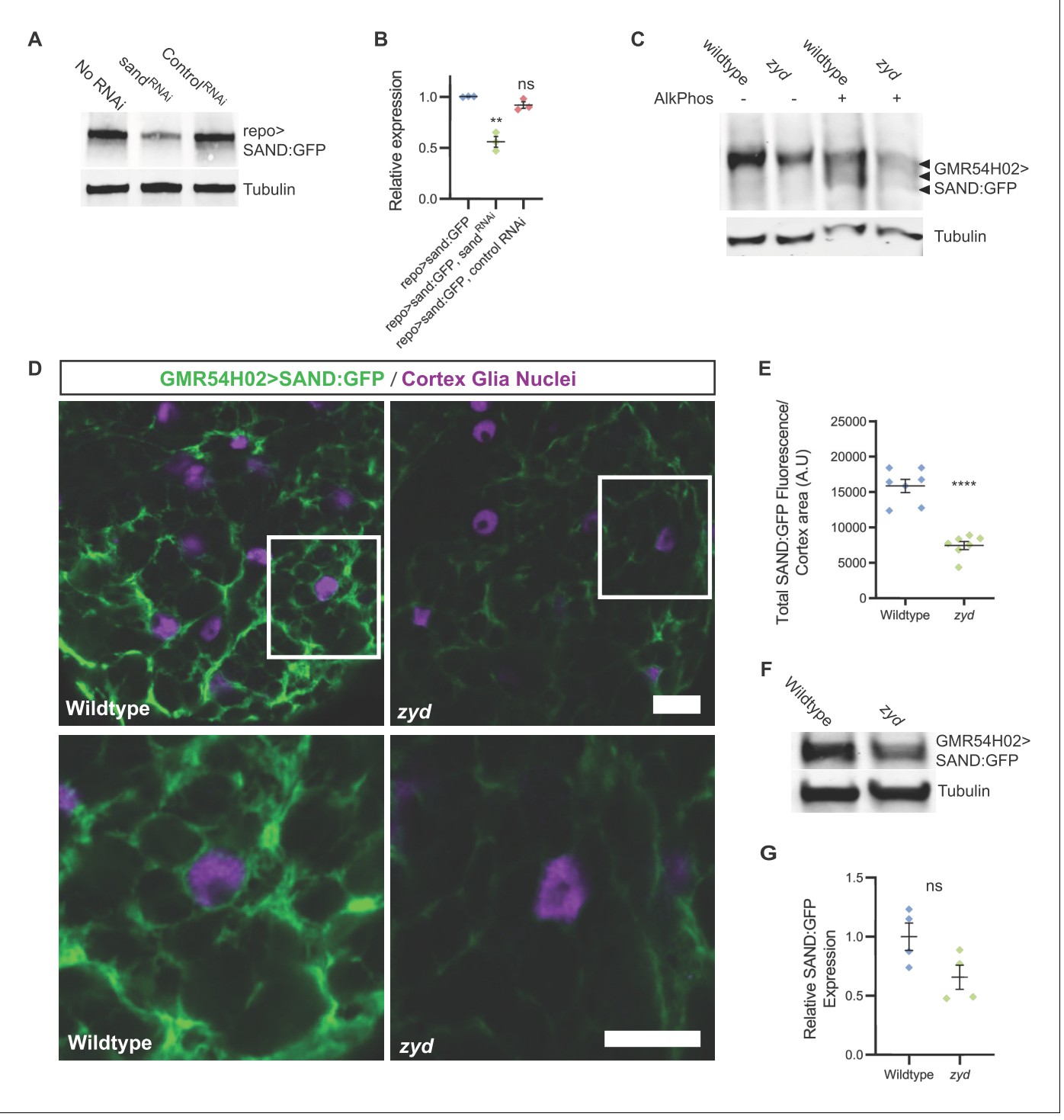

**Figure 6.** Sand protein levels and plasma membrane localization are reduced in *zyd* mutants. (A-B) Western blot analysis of sand[RNAi] knockdown of SAND:GFP (both sand:GFP and sand[RNAi] are driven with the pan glial driver, repo). SAND:GFP expression in sand[RNAi] knockdown is reduced by ~45% (p=0.0011), while the expression of a control RNAi does not significantly change SAND:GFP expression (N ≥ 2 experiments, three head extracts per sample). GFP signals in each experiment were normalized to control. (C) $Mn^{2+}$ phosphate binding tag (Phos-tag) gel electrophoresis analysis shows that SAND:GFP is not differentially phosphorylated in *zyd* relative to wildtype cortex glia, indicated by a single SAND:GFP band. Multiple bands were detected when samples were pre-treated with alkaline phosphatase, indicating SAND:GFP is phosphorylated on multiple phosphorylation sites in cortex glia (N = 3, five heads/sample). (D) Immunofluorescence of 3[rd] instar larval ventral nerve cords (VNCs). Cortex glial levels of SAND:GFP are reduced in *zyd* relative to wildtype (magenta: anti-repo, glial nuclei; green: anti-GFP, SAND:GFP; scale bars: 10 μm in upper panels, 5 μm in lower panels). (E) Average sand:GFP fluorescence in cortex glia of wildtype and *zyd*. SAND:GFP fluorescence is reduced by 53 ± 6.85% in *zyd* cortex glia.

*Figure 6 continued on next page*

*Figure 6 continued*

(n = 8 larvae/genotype). (F-G) Western blot analysis of SAND:GFP expression level in wildtype and *zyd* cortex glia (both driven with the cortex glial driver, GMR54H02-gal4). SAND:GFP expression in *zyd* cortex glia is reduced by ~34 ± 15% relative to wildtype (N ≥ 4 experiments, three heads/sample). Error bars are SEM, \*\*=P < 0.01, \*\*\*\*=P < 0.0001 Student's t-test.

DOI: https://doi.org/10.7554/eLife.44186.021

The following figure supplement is available for figure 6:

**Figure supplement 1.** Sand protein levels and plasma membrane localization are reduced in *zyd* mutants.

DOI: https://doi.org/10.7554/eLife.44186.022

expression level detected by western blot analysis was reduced by ~34 ± 15% in *zyd* cortex glia (*Figure 6F–G*). The reduction in expression was specific to sand, and not due to a general disruption in membrane homeostasis, as no difference was detected in membrane tethered GFP (mCD8:GFP) expressed in wildtype and *zyd* cortex glia (*Figure 1A*, *Figure 6—figure supplement 1B–C*). These results indicate that sand undergoes enhanced internalization and increased degradation in *zyd* relative to wildtype cortex glial cells.

If hyperactivity of CN leads to enhanced internalization of sand and subsequent seizures, interrupting cortex glial endocytosis should suppress *zyd* seizures. To test this model, we used cortex glial-specific RNAi to knock down genes involved in endocytosis and early endosomal processing and trafficking. Cortex glial knockdown of several essential endocytosis genes, including dynamin-1 and clathrin heavy and light chains, caused embryonic lethality (*Figure 2—figure supplement 1A*). In contrast, cortex glial knockdowns of Rab5, a Rab GTPase regulator of early endosome (EE) dynamics (*Dunst et al., 2015*; *Langemeyer et al., 2018*), and Endophilin A (EndoA), a BAR-domain protein involved in early stages of endocytosis (*Verstreken et al., 2002*), were found to be viable (*Figure 2—figure supplement 1A*) and to completely suppress *zyd* TS seizures (*Figure 7A*, *Figure 7—video 1*). A second, non-overlapping hairpin and a dominant negative (DN) construct for Rab5 (Rab5$^{DN}$) resulted in early larval lethality, likely due to more efficient Rab5 activity suppression. Cortex glial knockdown of Rab5 was previously shown to cause a morphological defect in which cortex glia fail to infiltrate the cortex and enwrap neuronal cell bodies (*Coutinho-Budd et al., 2017*). The loss of neuronal wrapping by cortex glia in Rab5$^{RNAi}$ might influence the glia-to-neuron signaling pathway that is activated in *zyd* animals to increase their seizure susceptibility. To test whether the rescue effect of Rab5 knockdown is due to an impairment in the structure of glial-neuronal contacts, we conditionally expressed Rab5$^{RNAi}$ and Rab5$^{DN}$ both in 3$^{rd}$ instar larvae and in adult cortex glia. Adult flies incubated for 16 hr at the restrictive temperature for gal80$^{ts}$ (31°C) to allow Rab5$^{RNAi}$ expression, showed a partial rescue in *zyd* seizures, with only ~25% of flies showing HS-induced seizures (*Figure 7B*). Adult flies and 3$^{rd}$ instar larvae conditionally expressing Rab5$^{DN}$ showed a partial rescue of *zyd* seizures with ~40% of animals displaying wildtype behavior (*Figure 7B*). These results suggest that Rab5, a master regulator of endocytosis and EE biogenesis, plays a key role for the pathway that is activated in *zyd* cortex glia to promote seizures. To assay if excess endocytosis secondary to CN hyperactivity disrupts membrane trafficking in cortex glia, we imaged endosomal compartments by over-expressing GFP-tagged Rab5 in cortex glial cells (with GMR54H02-gal4) in control and *zyd* animals. We found that large (>0.1 μm$^2$) Rab5-positive early endosomes accumulated in *zyd* cortex glia compared to controls (*Figure 7C,D*). Feeding *zyd* larvae the CN inhibitor, CsA (1 mM), restored the number of Rab5 compartments to control levels (*Figure 7D*). These results indicate CN hyperactivation secondary to elevated Ca$^{2+}$ levels in *zyd* mutants increases endocytosis and the formation and accumulation of early endosomes in cortex glia.

## Chronic inhibition of dynamin-mediated endocytosis rescues *zyd* seizures

Our previous analysis of *zyd* indicated that basal intracellular Ca$^{2+}$ is elevated in cortex glia, with Ca$^{2+}$ levels increasing more when *zyd* animals are heat-shocked (*Melom and Littleton, 2013*). The temperature-induced elevation in Ca$^{2+}$ could further enhance CN activity and endocytosis beyond that observed at rest. These data raise the question of whether the basal enhancement of endocytosis or the additional heat shock-induced Ca$^{2+}$ increase is the primary cause for seizure susceptibility in *zyd* mutants. To directly assay the role of endocytosis in seizure susceptibility in *zyd* flies, we

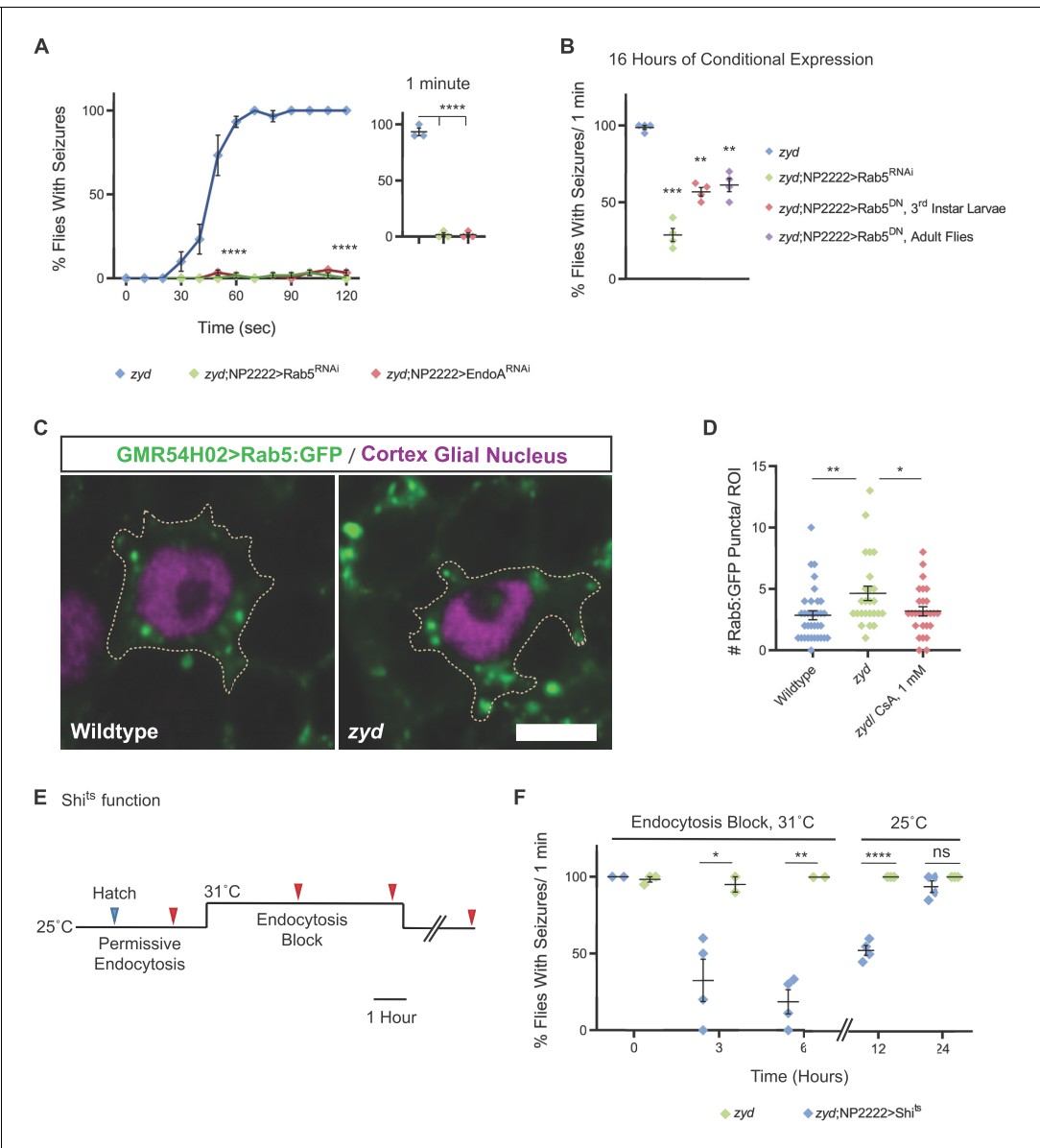

**Figure 7.** Cortex glial inhibition of endocytosis rescues *zyd* seizures. (**A-B**) Behavioral analysis of HS induced seizures. (**A**) Cortex glial knockdown of Rab5 and EndoA rescues seizures in the *zyd* mutant. Inset shows effects after 1 minute of HS (N = 3 groups of 20 flies/genotype, p<0.0001 at 1 minute and 2 minutes). (**B**) Cortex glial conditional overexpression of Rab5$^{RNAi}$ and dominant-negative Rab5 (Rab5$^{DN}$) using UAS/gal4/gal80$^{ts}$. Rearing adult flies at the restrictive temperature (>30°C) for gal80$^{ts}$ allows expression of Rab5$^{RNAi}$ or Rab5$^{DN}$ only in adults. A significant reduction in seizures was seen after 16 hr of incubation at the restrictive temperature for gal80$^{ts}$ (31°C), with only ~25% zyd/Rab5$^{RNAi}$ flies showing seizures (p=0.0006), and ~40% zyd/Rab5$^{DN}$ larvae (p=0.0015)/adult flies (p=0.0055) not showing seizures. (N = 3 groups of >10 flies/genotype). (**C**) Fluorescence images showing accumulation of Rab5::GFP puncta in *zyd* cortex glia relative to wildtype cortex glia. Rab5::GFP was expressed using a cortex glial-specific driver (GMR54H02-gal4; scale bar = 5 μm. n ≥ 5 animals/genotype). (**D**) Analysis of the number of large (>0.1 μm$^2$) Rab5::GFP puncta in wildtype and *zyd* cortex glia. The number of Rab5::GFP puncta in *zyd* cortex glia was increased relative to wildtype (average of 4.64 ± 0.58 and 2.85 ± 0.37 puncta/ROI respectively, p=0.0088). The number of large Rab5::GFP puncta in *zyd* treated with 1 mM CsA for 24 hr was decreased relative to *zyd* (average of 3.19 ± 0.37 puncta/ROI, p=0.0378; n ≥ 25 ROIs/3 animals/genotype/treatment). (**E-F**) Conditional inhibition of endocytosis by cortex glial overexpression of shi$^{ts}$. (**E**) Schematic representation of the experimental design. Adult *zyd*; NP2222 >shi$^{ts}$ male flies (>1 day old) were incubated at the shi$^{ts}$ restrictive temperature (31°C) for 3 or 6 hr and then tested for HS-induced seizures (red arrowheads, N = 3 groups of >15 flies/time point). (**F**) Behavioral analysis of HS induced seizures. Left: A significant reduction in seizures is observed in flies that were incubated at 31°C for 3 hr (p=0.0283) or 6 hr (p=0.013). (N = 3 groups of >15 flies/time point). Right: *zyd*;NP2222 >Shi$^{ts}$ flies seizures re-occur after removal from the Shi$^{ts}$ restrictive temperature (25°C, N = 4 groups of 10–15 animals/time point; 12 hr: p<0.0001). Error bars are SEM, *=P < 0.05, **=P < 0.01, ***=P < 0.001, ****=P < 0.0001, Student's t-test.

DOI: https://doi.org/10.7554/eLife.44186.023
*Figure 7 continued on next page*

*Figure 7 continued*

The following video is available for figure 7:

**Figure 7—video 1.** The response of zyd/NP2222>Rab5^RNAi flies to a 38.5°C heat-shock is shown.

DOI: https://doi.org/10.7554/eLife.44186.024

conditionally manipulated endocytosis by overexpressing a TS dominant-negative form of Dynamin-1 (Shi$^{ts}$) in *zyd* cortex glia. This mutant version of Dynamin has normal activity at room temperature and a dominant-negative function upon exposure to the non-permissive temperature (>29°C, *Figure 7E*). Acute inhibition of endocytosis by inactivation of Shi$^{ts}$ in cortex glia did not suppress *zyd* seizures, suggesting further enhancement of CN activity and endocytosis specifically during the heat shock is not likely to be the cause of the rapid-onset seizures observed in *zyd* mutants. Given the chronic enhancement in CN activity and endocytosis in *zyd* mutants demonstrated by enhanced CalexA signaling (*Figure 3*) and early endosome accumulation (*Figure 7C–D*), we hypothesized that inhibiting endocytosis prior to exposing animals to a heat shock might improve their phenotype by altering the plasma membrane protein content over longer timescales. We incubated *zyd*; NP2222>Shi$^{ts}$ flies at a non-permissive temperature for Shi$^{ts}$ (31°C) for either 3 or 6 hr, and then tested for heat shock-induced seizures at 38.5°C (*Figure 7E–F*). *Zyd* mutants alone do not undergo seizures at 31°C, nor does pre-incubation at 31°C alter the subsequent seizure phenotype observed at 38.5°C. In contrast, inhibition of endocytosis for 6 hr at 31 °C in *zyd* mutants co-expressing Shi$^{ts}$ suppressed the subsequent seizures observed during a 38.5°C heat shock in ~80% of animals (*Figure 6F*). A shorter 3 hr inhibition caused a less significant improvement in seizures. The seizure suppression observed after 6 hr of Dynamin inhibition was reversible, as adults tested 12 or 24 hr after return to room temperature regained the *zyd* seizure phenotype (*Figure 7F*, right). We conclude that chronic hyperactivation of CN and endocytosis caused by elevated basal Ca$^{2+}$ in *zyd* cortex glia is the primary cause for *zyd* seizures.

## Artificially increasing glial K$^+$ uptake rescues *zyd*- dependent seizures

Genetic analysis of *zyd* indicate the primary cause of seizure susceptibility is chronic enhancement in Ca$^{2+}$-dependent CN activity and subsequent increases in endocytosis in cortex glia. We hypothesize that this enhancement in endocytosis leads to increased internalization of sand, which in turn disrupts K$^+$ uptake and buffering by cortex glial cells during periods of intense neuronal activity (*Figure 8A*). If the cause of *zyd* seizures is a reduction in K$^+$ buffering, we predicted that increasing it by over-expression of sand will improve potassium homeostasis and reduce seizures. Indeed, pan-glial overexpression of sand:GFP partially suppressed *zyd* HS-induced seizures (*Figure 8C*). It is likely that a full rescue was not observed because the over-expressed channel is still internalized and degraded in *zyd* cortex glia beyond that found in controls (*Figure 6D–G*). To test this model further, we assayed if artificially increasing cortex glial K$^+$ uptake in *zyd* mutants by overexpressing another K$^+$ leak channel could suppress the seizure phenotype. Constitutive cortex glial overexpression of the open K$^+$ channel EKO (*White et al., 2001*) rescued vortex-induced seizures in ~75% of *zyd* mutants (*Figure 8B*). During a heat shock, cortex glial overexpression of EKO led to a dramatic change in the behavior of ~60% of *zyd* animals, showing partial recovery from the seizure phenotype to bottom dwelling and hypoactivity (*Figure 8C*). CPG recordings revealed that *zyd*;NP2222>EKO larvae regain rhythmic muscle activity at 38.5°C (*Figure 8D–E*), indicating cortex glial K$^+$ buffering is critical for neuronal excitability during states of intense excitation following heat shock or acute vortex. Together, these results indicate that enhanced CN-induced endocytosis lead to internalization and degradation of sand, impairment of cortex glial K$^+$ buffering and increased seizure susceptibility in *zyd* mutants.

## Discussion

Significant progress has been made in understanding glial-neuronal communication at synaptic and axonal contacts, but whether glia regulate neuronal function via signaling at somatic regions remains largely unknown. *Drosophila* cortex glia provide an ideal system to explore how glia regulate neuronal function at the soma, as they ensheath multiple neuronal cell bodies, but do not contact synapses (*Awasaki et al., 2008*). In this study, we took advantage of the *zydeco* (*zyd*) mutation in a Na$^+$/

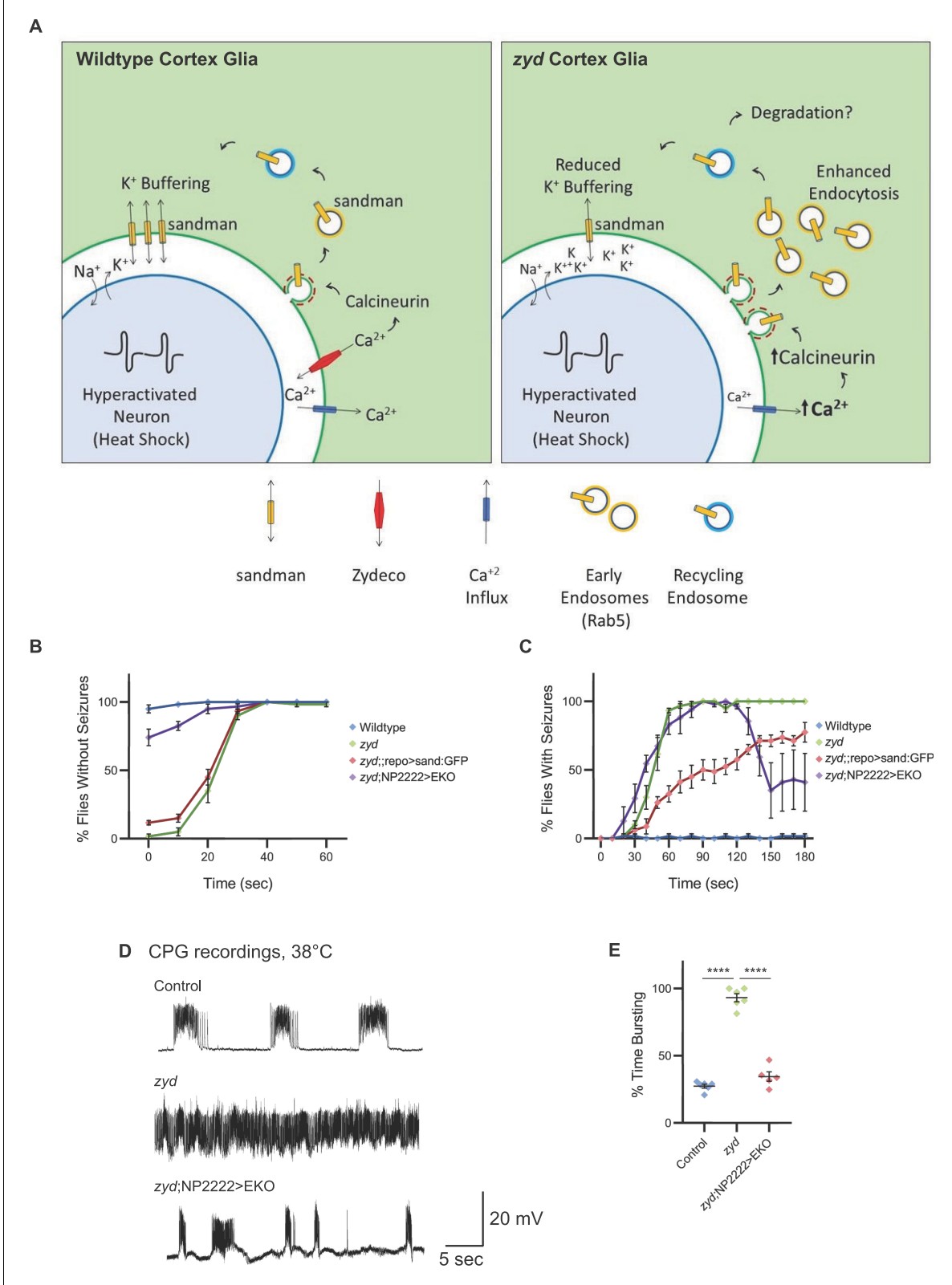

**Figure 8.** Enhancing glial K+ buffering by overexpressing a leak K+ channel rescues *zyd* seizures. (**A**) A model for *zyd* function in seizure susceptibility is depicted. In wildtype cortex glia (left), oscillatory Ca$^{2+}$ signaling maintains normal cortex glia-to-neuron communication and a balanced extracellular ionic environment. In *zyd* cortex glia (right), the basal elevation of Ca$^{2+}$ leads to hyperactivation of CN and enhanced endocytosis with accumulation of early endosomes. This disrupts the endo-exocytosis balance of the K$_{2P}$ leak channel sandman (and potentially other cortex glial membrane proteins)

*Figure 8 continued on next page*

*Figure 8 continued*

and impaired glial K$^+$ buffering. (B) Behavioral analysis of the recovery from vortex-induced seizures. Pan-glial over expression of SAND:GFP partially rescues *zyd* seizures (~15%), while cortex-glial overexpression of a genetically modified constitutively-open Shaker K$^+$ channel (termed EKO) rescues ~75% of *zyd* seizures. (N = 3 groups of 20 flies/genotype). (C) Behavioral analysis of HS induced seizures. Cortex glial overexpression of EKO lead to a dramatic change in the behavior of ~60% of *zyd* animals, showing partial recovery from the seizure phenotype to bottom dwelling and hypoactivity. (D-E) CPG recordings revealed that *zyd*;NP2222>EKO larvae regain rhythmic muscle activity. (D) Representative voltage traces of spontaneous CPG activity at 3$^{rd}$ instar larval muscle 6 at 38°C in wildtype, *zyd* and *zyd*;NP2222>EKO (*n* ≥ 3 preparations/genotype). Cortex-glial expression of EKO eliminates the continuous CPG seizures observed in *zyd* mutants. (E) Quantification of average bursting duration for CPG recordings of the indicated genotypes at 38°C. Error bars are SEM, ****=P < 0.0001, Student's t-test.

DOI: https://doi.org/10.7554/eLife.44186.025

Ca$^{2+}$/K$^+$ exchanger to explore how cortex glia regulate neuronal excitability. We found that elevation of basal Ca$^{2+}$ levels in cortex glia leads to hyperactivation of Ca$^{2+}$-CN dependent endocytosis. We showed that seizures in *zyd* mutants can be fully suppressed by either conditional inhibition of endocytosis or by pharmacologically reducing CN activity. Two well-characterized mechanisms by which glia regulate neuronal excitability and seizure susceptibility are neurotransmitter uptake via surface transporters and spatial K$^+$ buffering. Cortex glia do not contact synapses, making it unlikely they are exposed to neurotransmitters. Instead, cortex glial-knockdown of the two-pore K$^+$ channel (K$_{2P}$) sandman, the *Drosophila* homolog of TRESK/KCNK18, recapitulates *zyd* TS seizures, while sand:GFP expression level in *zyd* is reduced in *zyd* relative to wildtype cortex glia. These findings indicate impairment in K$^+$ buffering during hyperactivity in *zyd* mutants underlies the increased seizure susceptibility, providing an unexpected link between glial Ca$^{2+}$ signaling and K$^+$ buffering. We propose that *Drosophila* cortex glia regulate the expression levels of sand (and potentially other K$^+$ channels and/or plasma membrane proteins) on the cell membrane in a Ca$^{2+}$-regulated fashion in wildtype animals. When Ca$^{2+}$ is constitutively elevated in *zyd* mutants, this regulation is thrown out of balance. Together, these findings indicate that elevated Ca$^{2+}$ levels lead to hyperactivation of CN and elevated endocytosis, sand internalization, and impairment in K$^+$ buffering by cortex glia in *zyd* mutant animals (*Figure 8A*).

Different glial subtypes exhibit dynamic fluctuations in intracellular Ca$^{2+}$ *in vitro* (*Fatatis and Russell, 1992*; *Nett et al., 2002*) and in vivo (*Nimmerjahn et al., 2009*; *Porter and McCarthy, 1996*). These early discoveries led to the model that astrocytes can listen to and regulate neuronal and brain activity. Accumulated findings indicate that glial Ca$^{2+}$ signaling influences neuronal physiology on a rapid time scale. However, the signaling pathways underlying these astrocytic Ca$^{2+}$ transients and their relevance to brain activity are poorly defined and controversial (*Fiacco and McCarthy, 2018*; *Savtchouk and Volterra, 2018*). The *Drosophila zyd* mutation was identified in an unbiased genetic screen for behavioral mutants that triggered TS-dependent seizures, thus establishing the biological importance of the pathway before the gene mutation and cellular origin of the defect was known. The elevation in cortex glial Ca$^{2+}$ levels found in *zyd* mutants provides a mechanism to explore how this pathway influences neuronal excitability. We recently found that Ca$^{2+}$ elevation in astrocyte-like glia results in the rapid internalization of the astrocytic plasma membrane GABA transporter GAT and subsequent silencing of neuronal activity through elevation in synaptic GABA levels (*Zhang et al., 2017*). As such, Ca$^{2+}$-regulated endo/exocytic trafficking of neurotransmitter transporters and K$^+$ channels to and from the plasma membrane may represent a broadly used mechanism for linking glial Ca$^{2+}$ activity to the control of neuronal excitability at synapses and cell bodies, respectively.

Effective removal of K$^+$ from the extracellular space is vital for maintaining brain homeostasis and limits network hyperexcitability during normal brain function, as disruptions in K$^+$ clearance have been linked to several pathological conditions (*David et al., 2009*; *Leis et al., 2005*; *Somjen, 2002*). In addition to ion homeostasis, astrocytic K$^+$ buffering has been suggested as a mechanism for promoting hyperexcitability and engaging network activity (*Bellot-Saez et al., 2017*; *Wang et al., 2012*). Two mechanisms for astrocytic K$^+$ clearance have been identified, including net K$^+$ uptake (mediated by the Na$^+$/K$^+$ ATPase pump) and K$^+$ spatial buffering (via passive K$^+$ influx) (*Bellot-Saez et al., 2017*). While many studies indicate K$_{ir4.1}$, a weakly inward rectifying K$^+$ channel exclusively expressed in glial cells, is an important channel mediating astrocytic K$^+$ buffering, it is unlikely to be the only mechanism for glial K$^+$ clearance, as K$_{ir4.1}$ channels account for less than half of the K$^+$

buffering capacity of mature hippocampal astrocytes (*Ma et al., 2014*). Several studies have linked members of the $K_{2P}$ family, mainly TREK-1 and TWIK-1, to distinct aspects of astrocytic function (*Hwang et al., 2014*; *Olsen et al., 2015*; *Woo et al., 2012*). Our finding that cortex glial knockdown of the *Drosophila* KCNK18/TRESK $K_{2P}$ homolog sand triggers stress-induced seizures indicate glial $K_{2P}$ channels are also involved in $K^+$ homeostasis in the brain.

Approximately 50 million people worldwide have epilepsy, making it one of the most common neurological diseases globally (World Health Organization, 2018, http://www.who.int/en/). The traditional view assumes that epileptogenesis occurs exclusively in neurons. However, an astrocytic basis for epilepsy was proposed almost two decades ago (*Gómez-Gonzalo et al., 2010*; *Tashiro et al., 2002*; *Tian et al., 2005*). Although increased glial activity has been associated with abnormal neuronal excitability, the role of glia in the development and maintenance of seizures, and the exact pathway(s) by which abnormal glial $Ca^{2+}$ alter glia-to-neuron communication and neuronal excitability are poorly characterized (*Wetherington et al., 2008*). Beyond gliotransmission, astrocytes can regulate neuronal excitability through the uptake and redistribution of $K^+$ ions (*Bellot-Saez et al., 2017*; *Wang et al., 2012*) and neurotransmitters (*Rose et al., 2017*) from the extracellular space. In this study, we found that *Drosophila* cortex glial cells with elevated intracellular $Ca^{2+}$ impair $K^+$ buffering. The observation that several anti-epileptic drugs reduce glial $Ca^{2+}$ oscillations in vivo (*Tian et al., 2005*), together with the fact that ~ 30% of epilepsy patients are non-responders, suggest that pharmacologically targeting glial pathways might be a promising avenue for future drug development in the field. Several neuronal seizure mutants in *Drosophila* have already been demonstrated to respond to common human anti-epileptic drugs, indicating key mechanisms that regulate neuronal excitability are conserved from *Drosophila* to humans. Indeed, *zyd*-induced seizures can be rescued when animals are fed a CN inhibitor (*Figure 4*), indicating pharmacological targeting of the CN pathway can improve the outcome of glial-derived seizures. Prior studies have also shown improvement following treatment with the CN inhibitor FK506 in a rodent kindling model (*Moia et al., 1994*; *Moriwaki et al., 1996*), suggesting CN activity may regulate epileptogenesis in both *Drosophila* and mammalian models. Further characterization of how glia detect, respond, and actively shape neuronal excitability is critical to our understanding of neuronal communication and future development of new treatments for epilepsy.

# Materials and methods

**Key resources table**

| Reagent type (species) or resource | Designation | Source or reference | Identifiers | Additional information |
|---|---|---|---|---|
| Genetic reagent (*D. melanogaster*) | w1118 | | | |
| Genetic reagent (*D. melanogaster*) | *zyd[1]* | *Melom and Littleton, 2013* | | *zyd* |
| Genetic reagent (*D. melanogaster*) | repo-gal4 | *Lee and Jones, 2005* | | |
| Genetic reagent (*D. melanogaster*) | NP2222-gal4 | *Hayashi et al., 2002* | RRID:DGGR_112830 | |
| Genetic reagent (*D. melanogaster*) | GMR5H02-gal4 | | RRID:BDSC_45784 | |
| Genetic reagent (*D. melanogaster*) | GMRH02-lexA | | | Gift of Gerald M Rubin |
| Genetic reagent (*D. melanogaster*) | UAS-cam-RNAi | | RRID:BDSC_34609 | Cam[RNAi] |
| Genetic reagent (*D. melanogaster*) | UAS-CanB-RNAi | | RRID:BDSC_27307 | CanB[RNAi#1] |
| Genetic reagent (*D. melanogaster*) | UAS-CanB-RNAi | | VDRC 21611 RRID:FlyBase_FBst0454139 | CanB[RNAi#2] |

*Continued on next page*

*Continued*

| Reagent type (species) or resource | Designation | Source or reference | Identifiers | Additional information |
|---|---|---|---|---|
| Genetic reagent (D. melanogaster) | UAS-CanB-RNAi | | VDRC 52390 RRID:FlyBase_FBst0469806 | CanB[RNAi#3] |
| Genetic reagent (D. melanogaster) | UAS-CanB2-RNAi | | RRID:BDSC_27270 | CanB2[RNAi#1] |
| Genetic reagent (D. melanogaster) | UAS-CanB2-RNAi | | VDRC 104370 | CanB2[RNAi#2] |
| Genetic reagent (D. melanogaster) | UAS-CanB2-RNAi | | VDRC 28764 RRID:FlyBase_FBst0457632 | CanB2[RNAi#3] |
| Genetic reagent (D. melanogaster) | UAS-CanB2-RNAi | | RRID:BDSC_38971 | CanB2[RNAi#4] |
| Genetic reagent (D. melanogaster) | UAS-Pp2B-14D-RNAi | | RRID:BDSC_25929 | Pp2B-14D[RNAi#1] |
| Genetic reagent (D. melanogaster) | UAS-Pp2B-14D-RNAi | | RRID:BDSC_40872 | Pp2B-14D[RNAi#2] |
| Genetic reagent (D. melanogaster) | UAS-CanA-14F-RNAi | | RRID:BDSC_38966 | CanA-14F[RNAi#1] |
| Genetic reagent (D. melanogaster) | UAS-CanA-14F-RNAi | | VDRC 30105 RRID:FlyBase_FBst0458337 | CanA-14F[RNAi#2] |
| Genetic reagent (D. melanogaster) | UAS-cac-RNAi | | VDRC 104168 | cac[RNAi] |
| Genetic reagent (D. melanogaster) | UAS-sand-RNAi | | VDRC 47977 RRID:FlyBase_FBst0467653 | sand[RNAi#1] |
| Genetic reagent (D. melanogaster) | UAS-sand-RNAi | | RRID:BDSC_25853 | sand[RNAi#2] |
| Genetic reagent (D. melanogaster) | UAS-Rab5-RNAi | | RRID:BDSC_34832 | Rab5[RNAi] |
| Genetic reagent (D. melanogaster) | UAS-Rab5.S43N | | RRID:BDSC_42703 RRID:BDSC_42704 | Rab5[DN] |
| Genetic reagent (D. melanogaster) | UAS-zyd-RNAi | | VDRC 40987 RRID:FlyBase_FBst0463881 | zyd[RNAi] |
| Genetic reagent (D. melanogaster) | UAS-EKO[+] | *White et al., 2001* | RRID:BDSC_40973 | |
| Genetic reagent (D. melanogaster) | CalexA | *Masuyama et al., 2012* | RRID:BDSC_66542 | |
| Genetic reagent (D. melanogaster) | Tub-gal80[ts] | | RRID:BDSC_7018 RRID:BDSC_7019 | |
| Genetic reagent (D. melanogaster) | UAS- Pp2B-14F[H217Q] | *Takeo et al., 2012* | RRID:DGGR_109869 | |
| Genetic reagent (D. melanogaster) | CanB2[KO] | *Nakai et al., 2011* | | Gift of Toshiro Aigaki |
| Genetic reagent (D. melanogaster) | UASc-sand:GFP | Generated in this study | | |
| Antibody | Rabbit polyclonal anti-GFP | ThermoFisher | #A11122 RRID:AB_221569 | Western blot, 1:5000 |
| Antibody | Mouse monoclonal anti-syx1A | DSHB | #8C3 RRID:AB_528484 | Western blot, 1:500 |
| Antibody | Mouse monoclonal anti-Tubulin | Sigma Aldrich | #T5168 RRID:AB_477579 | Western blot, 1:1,000,000 |
| Antibody | Rabbit polyclonal anti-cleaved-*Drosophila* DCP1 | Cell Signaling | #9578 RRID:AB_2721060 | Western blot, 1:250 |

*Continued on next page*

*Continued*

| Reagent type (species) or resource | Designation | Source or reference | Identifiers | Additional information |
|---|---|---|---|---|
| Antibody | Mouse monoclonal anti-repo | DSHB | #8D12 RRID:AB_528448 | IF, 1:25 |
| Antibody | Rat monoclonal anti-elav | DSHB | #7E8A RRID:AB_2800446 | IF, 1:50 |
| Antibody | Rabbit polyclonal antiGFP-488 | Invitrogen | #A21311 RRID:AB_221477 | IF, 1:500 |
| Antibody | Goat polyclonal anti-Mouse405 | Life technologies | #A31553 RRID:AB_221604 | IF, 1:3000 |
| Antibody | Goat polyclonal anti-Rat555 | Invitrogen | #A21434 RRID:AB_2535855 | IF, 1:3000 |
| Antibody | IRDye680LT Goat anti-Mouse IgG Secondary Antibody | LI-COR | #926–68020 RRID:AB_10706161 | Western blot, 1:3000 |
| Antibody | IRDye800CW Goat anti-Rabbit IgG Secondary Antibody | LI-COR | #926–32211 RRID:AB_621843 | Western blot, 1:3000 |
| Chemical compound, drug | CyclosporinA | Sigma Aldrich | #30024 | 1 mM |
| Chemical compound, drug | FK506 | InvivoGen | tlrl-fk5 | 1 mM |
| Enzyme | Alkaline Phospatase | Promega | #M282A | |

*For a complete list of all RNAi stocks used in this study, see **Supplementary file 2**.

### *Drosophila* genetics and molecular biology

Flies were cultured on standard medium at 22°C unless otherwise noted. *zydeco* (*zyd¹*, here designated as *zyd*) mutants were generated by ethane methyl sulfonate (EMS) mutagenesis and identified in a screen for temperature-sensitive (TS) behavioral phenotypes (*Guan et al., 2005*). The UAS/gal4 and LexAop/LexA systems were used to drive transgenes in glia, including repo-gal4 (*Lee and Jones, 2005*), a pan-glial driver; NP2222-gal4, a cortex-glial specific driver; GMR54H02-gal4, expressed in a smaller set of cortex glial cells; and alrm-gal4, an astrocyte-like glial cell specific driver. The *UAS-dsRNAi* flies used in the study were obtained from the VDRC (Vienna, Austria) or the TRiP collection (Bloomington *Drosophila* Stock Center, Indiana University, Bloomington, IN, USA). All screened stocks are listed in supplementary material (*Supplementary file 2*). UAS-*myrG-CaMP6s* was constructed by replacing GCaMP5 in the previously described myrGCaMP5 transgenic construct (*Melom and Littleton, 2013*). UAS-sand:GFP was constructed by fusing the ORF of sand (*Drosophila melanogaster* sandman (sand), mRNA, NM_136505) to eGFP and inserted into pBID-UASc between restriction sites EcoRI and XbaI (Epoch Life Science, Inc). Transgenic flies were obtained by standard germline injection (BestGene Inc). For all experiments described, only male larva and adults were used, unless otherwise noted. In RNAi experiments, the animals also had the UAS-dicer2 transgenic element on the X chromosome to enhance RNAi efficiency. For survival assays, embryos were collected in groups of ~50 and transferred to fresh vials (n = 3). 3rd instar larvae and/or pupae were counted. Survival rate (SR) was calculated as:

$$SR = \frac{N_{live\ 3rd\ instar\ animals}}{N_{embryos}}$$

For conditional expression using Tub-gal80ts (*Figure 1E*, *2F*, *7B*), animals of the designated genotype were reared at 22°C with gal80 suppressing gal4-driven transgene expression (zyd^RNAi, CanB2^RNAi and Rab5^DN and respectively). 3rd instar larvae or adult flies were then transferred to a 31°C incubator to inactivate gal80 and allow gal4 expression or knockdown for the indicated period. For inhibiting transgene expression in cortex glia (*Figure 5A*) GMR54H02-lexA (a kind gift from

Gerald Rubin collection) was used to express gal80 from LexAop-gal80. For inhibiting transgene expression specifically in neurons (*Figure 2E*, *Figure 2—figure supplement 1D*), C155-gal80 was used.

## Behavioral analysis

All experiments were performed using groups of ~10–20 males.

### Temperature-sensitive seizures/*zyd* modifier screen

Adult males aged 1–2 days were transferred in groups of ~10–20 flies (n ≥ 3, total # of flies tested in all assays was always >40) into preheated vials in a water bath held at the indicated temperature with a precision of 0.1°C. Seizures were defined as the condition in which the animal lies incapacitated on its back or side with legs and wings contracting vigorously (*Melom and Littleton, 2013*; *Figure 1—video 1*). For screening purposes, only flies that showed normal wildtype-like behavior (i.e. walking up and down on vial walls, *Figure 1—video 1*, *Figure 2—video 4* and *5*) during heatshock were counted as successful rescue. To analyze behavior in a more detailed manner, we characterized four behavioral phenotypes: wall climbing (flies are climbing on vial walls), bottom dwelling (flies are on the bottom of the vial, standing/walking without seizures), partial seizures (flies are on the bottom of the vial, seizing most of the time) and complete seizures (flies are constantly lying on their side or back with legs twitching). For assaying seizures in larvae, 3rd instar males were gently washed with PBS and transferred to 1% agarose plates heated to 38°C using a temperature-controlled stage (*Melom and Littleton, 2013*). Larval seizures were defined as continuous, unpatterned contraction of the body wall muscles that prevented normal crawling behavior (*Melom and Littleton, 2013*). For determining seizure temperature threshold, groups of 10 animals were heat-shocked to the indicated temperature (either 37.5, 38, 38.5 or 39°C). Threshold was defined as the temperature in which > 50% of the animal were seizing after 1 minute.

### Bang sensitivity

Adult male flies in groups of ~10–20 males (n = 3) were assayed 1–2 days post-eclosion. Flies were transferred into empty vials and allowed to rest for 1–2 hr. Vials were vortexed at maximum speed for 10 seconds, and the number of flies that were upright and mobile was counted at 10 s intervals.

### Light avoidance

These assays were performed using protocols described previously following minor modifications. Briefly, pools of ~20 3rd instar larvae (108–120 hr after egg laying) were allowed to move freely for 5 minutes on Petri dishes with settings for the phototaxis assay (Petri dish lids were divided into quadrants, and two of these were blackened to create dark environment). The number of larvae in light versus dark quadrants was then scored (n = 4). Response indices (RI) were calculated as:

$$RI = \frac{N_{Dark}}{total}$$

### Activity monitoring using the MB5 system

Adult flies activity was assayed using the multi-beam system (MB5, TriKinetics) as previously described (*Green et al., 2015*; *McParland et al., 2015*). Briefly, individual males aged 1–3 days were inserted into 5 mm ×80 mm glass pyrex tubes. Activity was recorded following a 20–30 minutes acclimation period. Throughout each experiment, flies were housed in a temperature- and light-controlled incubator (25°C, ~40–60% humidity). Post-acquisition activity analysis was performed using Excel to calculate activity level across 1 minute time bins (each experimental run contained eight control animals and eight experimental animals, n ≥ 3).

### Gentle touch assay

3rd instar male larvae (108–120 hr after egg laying) were touched on the thoracic segments with a hair during forward locomotion. No response, a stop, head retraction and turn were grouped into type I responses, and initiation of at least one single full body retraction or multiple full body retractions were categorized as type II reversal responses. Results were grouped to 20 males per assay (n = 3).

## Electrophysiology

Intracellular recordings of wandering 3rd instar male larvae were performed in HL3.1 saline (in mm: 70 NaCl, 5 KCl, 4 MgCl$_2$, 0.2 CaCl$_2$, 10 NaHCO$_3$, 5 Trehalose, 115 sucrose, 5 HEPES-NaOH, pH 7.2) containing 1.5 mm Ca$^{2+}$ using an Axoclamp 2B amplifier (Molecular Devices) at muscle fiber 6/7 of segments A3-A5. For recording the output of the central pattern generator, the CNS and motor neurons were left intact. Temperature was controlled with a Peltier heating device and continually monitored with a microprobe thermometer. Giant fiber recordings were performed and seizure thresholds were defined as previously described (*Howlett and Tanouye, 2013*; *Pavlidis and Tanouye, 1995*). Shortly, seizure-like activity was defined as uncontrolled, high-frequency (>100 Hz) motoneuron activity evoked by HFS stimulation and recorded in the DLM (*Kuebler and Tanouye, 2000*). Seizure threshold was the lowest HFS voltage that evoked seizure-like activity.

## In vivo Ca$^{2+}$ imaging

UAS-*myrGCaMP6s* was expressed in glia using the drivers described above. 2nd instar male larvae were washed with PBS and placed on a glass slide with a small amount of Halocarbon oil #700 (Lab-Scientific). Larvae were turned ventral side up and gently pressed with a coverslip and a small iron ring to inhibit movement. Images were acquired with a PerkinElmer Ultraview Vox spinning disk confocal microscope and a high-speed EM CCD camera at 8–12 Hz with a 40 × 1.3 NA oil-immersion objective using Volocity Software. Single optical planes within the ventral cortex of the ventral nerve cord (VNC) were imaged in the dense cortical glial region immediately below the surface glial sheath. Average myrGCaMP6s signal in cortex glia was quantified in the central abdominal neuromeres of the VNC within a manually selected ROI excluding the midline glia. Ca$^{2+}$ oscillations were counted within the first minute of imaging at room temperature, and then normalized to the ROI area.

## Drug feeding

Cyclosporin-A (CsA, Sigma Aldrich) or FK506 (InvivoGen) were dissolved in DMSO to a final concentration of 20 mM. The feeding solution included 5% yeast and 5% sucrose in water. Adult males less than 1 day old were starved for 6 hours and then transferred to a vial containing a strip of Wattman paper soaked in feeding solution containing the designated concentration of CsA/FK506 or DMSO as control. Flies were behaviorally tested following 6, 12 or 24 hours of drug feeding.

## Immunostaining, Western blot and Phos-tag Analysis

For immunostaining, dissected 3rd instar male larvae were fixed with cold 4% paraformaldehyde in HL3.1 buffer for 45 minutes. Antibodies were used at the following dilutions: mouse anti-repo (8D12 Developmental Studies Hybridoma Bank), 1:50; rat anti-ELAV (7E8A10, Developmental Studies Hybridoma Bank), 1:50; GFP Rabbit IgG, Alexa Fluor 488 Conjugate (Thermo Fisher, 1:500); Goat anti mouse Alexa Fluor 405 Conjugate (Life technologies, 1:2000) and Goat anti-rat Alexa Fluor 555 Conjugate (Invitrogen, 1:2000). Larvae were mounted in VECTASHIELD (Vector Labs) and imaged on a Zeiss LSM800 confocal microscope with ZEN software (Carl Zeiss MicroImaging) with oil-immersion 63/1.4 NA objectives. Morphological analysis and quantification were performed using Imaris software. Rab5::GFP was expressed specifically in cortex glia using GMR54H02-gal4 driver. GFP puncta (>0.1 μm$^2$) were detected automatically within a set circular ROI (with r = 5 μm, centroid in the center of the repo positive nucleus) using Volocity software. Western blotting of adult whole-head and larval brain lysates was performed using standard laboratory procedure. Nitrocellulose membranes were probed with rabbit anti-cleaved DCP1 (Cell Signaling, 1:1000) and rabbit anti-GFP (Abcam, 1:500). Equal loading was assayed using mouse anti-syx1A (1:1000). Primary antibodies were detected with Alexa Fluor 680-conjugated and 800-conjugated anti-rabbit and anti-mouse (Invitrogen, 1:3000). Western blots were visualized with an Odyssey infrared scanner (Li-Cor). For phosphorylation analysis of SAND:GFP, fly's heads were homogenized in lysis buffer (50 mM HEPES, pH = 7.15, 1% NP-40, 150 mM NaCl, 3 mM MgCl$_2$, 10% glycerol) containing phosphatase inhibitor (Halt Phosphatase Inhibitor Cocktail, Thermo Scientific, #78420, 1:100), on ice. Control samples were treated with Alkaline phosphatase for 20 minutes on ice. Phos-tag gels (7.5%, Wako Chemicals, #198–17981) were used according to the manual, following by standard laboratory procedure.

## Electron microscopy (EM)

Third-instar wildtype and *zyd* larvae were dissected in $Ca^{2+}$-free solution and fixed in 4% formaldehyde, and 0.1 m sodium cacodylate at 4°C overnight. After washing in 0.1 m sodium cacodylate and 0.1 m sucrose, samples were postfixed for 1 hr in 1% osmium tetroxide, dehydrated through a graded series of ethanol and then acetone, and embedded in epoxy resin (Embed 812; Electron Microscopy Sciences). Thin sections (40–50 nm) were collected on Formvar/carbon-coated copper slot grids and contrasted with lead citrate. Sections were imaged on an electron microscope (Tecnai G2 Spirit; FEI) equipped with a charge-coupled device camera (Advanced Microscopy Techniques).

## Statistical analysis

No statistical methods were used to predetermine sample size. All *n* numbers represent biological replicates. Data were pooled from 2 to 3 independent experiments. Immunofluorescence experiments ($Ca^{2+}$ imaging, CalexA expression and Rab5 puncta characterization) were randomized and blinded. P values are represented as *=$P < 0.05$, **=$P < 0.01$, ***=$P < 0.001$, ****=$P < 0.0001$. $p<0.05$ was considered significant. All data are expressed as mean ± SEM.

## Acknowledgements

This work was supported by NIH grants NS40296 and MH104536 to JTL. We thank the Bloomington *Drosophila* Stock Center (NIH P40OD018537), the Vienna *Drosophila* RNAi Center, the Harvard TriP Project, the KYOTO Stock Center, Marc Freeman (Vollum Institute), Gerald Rubin (Janelia Research Campus) and Toshiro Aigaki (Tokyo Metropolitan University) for providing *Drosophila* strains, the Developmental Studies Hybridoma Bank for antisera, Mark Tanouye for help with GF recordings, and members of the Littleton lab for helpful discussions and comments on the manuscript.

## Additional information

### Funding

| Funder | Grant reference number | Author |
|---|---|---|
| National Institutes of Health | NS40296 | J Troy Littleton |
| National Institutes of Health | MH104536 | J Troy Littleton |

The funders had no role in study design, data collection and interpretation, or the decision to submit the work for publication.

### Author contributions

Shirley Weiss, Conceptualization, Data curation, Formal analysis, Investigation, Writing—original draft, Writing—review and editing; Jan E Melom, Kiel G Ormerod, Yao V Zhang, Data curation; J Troy Littleton, Conceptualization, Supervision, Writing—original draft, Writing—review and editing

### Author ORCIDs

Shirley Weiss (ID) http://orcid.org/0000-0002-1006-349X
J Troy Littleton (ID) http://orcid.org/0000-0001-5576-2887

### Decision letter and Author response

Decision letter https://doi.org/10.7554/eLife.44186.031
Author response https://doi.org/10.7554/eLife.44186.032

## Additional files

### Supplementary files

• Supplementary file 1. Summary of *zyd* suppressor/enhancer RNAi screen. A table summarizing the results of the *zyd* suppressor/enhancer RNAi screen *performed in this study.*
DOI: https://doi.org/10.7554/eLife.44186.026

• Supplementary file 2. Summary of all RNAi hairpins used in this study. A table summarizing all the RNAi hairpins used for the *zyd* suppressor/enhancer and CN target RNAi screens.
DOI: https://doi.org/10.7554/eLife.44186.027

• Supplementary file 3. Summary of RNAi hairpins used in the CN target screen. A table summarizing all the RNAi hairpins used for the CN target RNAi screens.
DOI: https://doi.org/10.7554/eLife.44186.028

• Transparent reporting form
DOI: https://doi.org/10.7554/eLife.44186.029

### Data availability

All data generated or analyzed during this study are included in the manuscript and supporting files.

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
