## [Decision Letter]

[Editors’ note: this article was originally rejected after discussions between the reviewers, but the authors were invited to resubmit after an appeal against the decision.]

Thank you for submitting your work entitled "Glial Ca^2+^ Signaling Links Endocytosis to K^+^ Buffering Around Neuronal Somas to Regulate Excitability" for consideration by *eLife*. Your article has been reviewed by three peer reviewers, including Graeme W Davis as the Reviewing Editor and Reviewer #1, and the evaluation has been overseen by a Senior Editor. The following individual involved in review of your submission has agreed to reveal their identity: Kai Zinn (Reviewer #3).

Our decision has been reached after consultation between the reviewers. Based on these discussions and the individual reviews below, we regret to inform you that your work will not be considered further for publication in *eLife*.

Summary:

All three reviewers were enthusiastic about the topic, the large extent of data and the use of unbiased forward genetic mechanisms to explore and provide mechanistic insight into the biology of uncontrolled activity that may be referred to as seizure in *Drosophila*. However, all of the reviewers also agree that the study would greatly benefit from a few additional experiments that would likely take more than 2 months, but thought that the study is really worth this additional effort. Our particular concern was the ultimate reliance on RNAi as an experimental tool to manipulated gene expression throughout the work. The early experiments on Zyd are backed up by mutant analyses. It was noted, for example, that the RNAi-based targeting of calcineurin used multiple RNAi that appeared to be overlapping based upon information from Flybase, and to have similarity with other members of the gene family. This could be easily addressed with a classical mutation. The work on Sandman, in particular, would benefit from a classical mutation, although it is acknowledged that prior publications in prominent journals have relied exclusively on the RNAi of Sandman. There were difference in opinion regarding how necessary the endocytosis data are for the entire manuscript given that it would be difficult or impossible to create reagents to directly follow the trafficking of the potassium channel, as a proposed mechanism. In conclusion, the reviewers consider this a paper that could be exceptional and appropriate for *eLife* with additional work that is likely to require a time commitment longer than allowed by *eLife* policy.

*Reviewer #1:*

The paper from the laboratory of Troy Littleton presents an extensive data set that extends upon prior work on the zyd gene. The authors perform a reasonably large screen of ~850 genes, looking for genetic interactions with zyd that rescue or enhance or otherwise modify a phenotype of enhanced neural activity, referred to as seizure. The paper as a whole includes a large amount of data, and an unbiased approach toward understanding the control of neuronal activity, specifically emphasizing the role of somatic glia. Here, the authors make a nice point of emphasis that the glia being studied are largely restricted to the somatic and initial processes of the cells, being largely separate from the synaptic neuropil. Thus, the point of regulation is somewhat anatomically defined for these cells. This is novel and interesting. The authors go on to present a compelling picture of how zyd functions through calcineurin and 2P potassium channels. This is a significant advance, particularly given the importance of glia in the control of neural function, but the general lack of mechanism in the broader literature. There are some specific weaknesses, particularly relevant that the final stages of pathway dissection that can either be addressed with new experiments or removed from the manuscript. Ultimately, the study provides a compelling and significant advance, working at a fundamental level of signaling discovery and the data should appeal to a wide audience.

1) Physiological data traces are shown as representative throughout the paper, without quantification. Thus, it is difficult to assess phenotype variance in wild type and mutations. It should be straightforward to quantify power of burst frequencies or some such measure and report for wild type and mutant in at least a few of these figures (Figure 2C, Figure 5C, Figure 7D).

2) The data sets generally are presented without wild type or genotype controls. These need not be added to every figure panel, but greater presentation of wild type data is important. This is essential in Figure 1A and Figure 7D at a minimum, but would be nice to see elsewhere as well.

3) Video of the seizure phenotype would be helpful. Most quantification is of 'percent flies with seizures'. This is an experimenter-based metric and seems solid given that the phenotypes are generally 100%. But, greater confidence in the data would be generated by showing how robust and obvious the phenotype actually is to the naïve observer.

4) The weakest point of the data set is the connection to endocytosis of 2P channel. Ideally, the channel would be monitored and shown to have altered trafficking. This is likely impossible given the current tools. However, examination of Rab5 itself could be indicative of many different cell biological changes. I am not sure what to suggest. It might be sufficient to remove these data and simply cite the two possible mechanisms of action for the signaling system rather than presenting a strong conclusion at this point. There is sufficient data in the manuscript without this experiment. On the other hand, genetic interaction with Hrs was documented in the secondary screen and Hrs is a late endosomal protein. An alternative to emphasize the endosomal interactions, with addition of new data, without proposing a direct mechanism of action.

5) The other weak point is proposed 'rescue' by EKO expression. Over-expression of potassium leak will clamp a cell at potassium equilibrium and prevent activity. I am not sure that this can be considered to be specifically 'reversing' the seizure phenotype in Figure 7. It might be nice data to incorporate into Figure 1 as an experiment to simply suppress excess activity?

6) Please provide primary (sample data) for calcium transients in Figure 2 and 5.

7) The data in Figure 5 presents the phenotypic similarity of *zyd* and sand mutants. The authors conclude that they are part of the same process. The problem is that both mutant conditions are fully penetrant, so putting them together is fairly inconclusive. The authors argue that seizures do not get worse in the double mutant condition, but there does not seem to be any quantitative way to assess a seizure getting worse, and the frequency is maxed out already. The authors could text a double heterozygous condition. They authors could text the double mutants and look for changes in threshold or over-express sand in the *zyd* mutant background. Something along these lines should be performed to strengthen the desired conclusion.

8) In Figure 1D, the authors present recordings in the giant fiber system, which are presumed to be recordings from muscle output, as is standard in the field (though not specified in the text). The authors refer to a seizure, but do not define what level of change in activity represents a 'seizure'. Second, and just as important, they refer to this change as indicative of altered intrinsic excitability. This conclusion is not appropriate without directly recording from central neurons since intrinsic excitability has a precise definition. Better definition of criteria and an different interpretation, more closely tied to the date, are necessary here.

*Reviewer #2:*

The group of T. Littleton had previously reported that in *Drosophila* cortex glia, the Na^+^Ca^2+^K^+^ exchanger (NCKX) Zydeco maintains low Ca^2+^ levels to control neuronal excitability. In the absence of Zydeco, Ca^2+^ oscillations in cortex glia are abolished and zydeco mutants exhibit increased susceptibility to stress-induced seizures. Since glial expression of calmodulin is required for seizure induction, it had been suggested that a Ca^2+^ /calmodulin-dependent signaling pathway controls glial-neuronal communication. Here the authors follow up on these results by first doing a small in vivo modifier screen. This resulted in the identification of the phosphatase Calcineurin which as the authors propose controls endocytosis in cortex glia to regulate the expression of a K2P channel Sandman. Thereby the authors present an interesting link between glial Ca^2+^ signaling and glial K^+^ buffering activity.

Overall this is a very interesting report, highly significant to a broad readership. However, at many places the conclusions of the authors are not fully supported by the data.

Overall the paper would greatly benefit from shortening.

*zydeco* mutants are homozygous viable, but suppression of *zydeco* expression by RNAi results in lethality. To me this suggests that RNAi is not the best way to assay this gene and therefore other approaches should be used to discriminate between a developmental or a physiological role of *zydeco* (e.g. conditional rescue in cortex glia instead of conditional RNAi based knockdown).

The authors state that the *zydeco* seizure phenotype is not due to morphological abnormalities of the cortex glia and show a rather low-resolution confocal image. Use single cell analysis (MARCM, Brainbow) and TEM.

The suppressor/enhancer screen is interesting, but the authors should present all gene names. For all positive candidates, at least one additional RNAi line or better a mutant should be used. This is mentioned in the text, but I saw no information on the results. Please include this in the table as well (e.g. how many of the hits were independently verified?).

The screen identified that knockdown of the phosphatase Calcineurin B2 rescued stress induced seizures in *zydeco* mutants. A similar rescue was observed using three additional non-overlapping RNAi constructs. However, according to flybase the RNAi constructs used in this experiment are indeed overlapping (and have quite some sequence homology to CanB). A mutant analysis should be done!

This is also true for CanA-14F and Pp2B14D, where mutants and P-insertions are available.

Increased Ca^2+^ levels in *zydeco* mutants result in increased Calcineurin activity. The image provided in Figure 3 does not match the western and quantification data.

Pharmacological blockage of the Calcineurin pathway rescues stress-induced seizures in *zydeco* mutants. The statement that this treatment affects the Calcineurin pathway in glial cells is not supported by the data and should be removed.

In a next step the authors study putative Calcineurin target proteins including the K^+^ channel Sandman. Again, the analysis is on the level of RNAi induced knockdown only and should include a mutant study. A crucial aspect of the paper is how Sandman function can be coupled to Ca^2+^ levels controlled by Zydeco. The mammalian Sandman orthologue is regulated by phosphorylation and two of the four phosphorylated tyrosine residues are conserved. Inhibition of kinases possibly phosphorylating Sandman gave negative results which does not allow any conclusions. Specific sandman mutants would help.

Throughout the text the authors frequently compare *Drosophila* cortex glia with mammalian glia – mostly astrocytes. This is in particular relevant in the context of K^+^ buffering. However, insect neurons are generally unipolar which means that the neuronal cell body is not involved in signal propagation as it is in mammalian neurons. The question that needs to be addressed is therefore where the different exchangers / channels (Zydeco or Sandman) are expressed within the cell.

The authors claim that endocytosis is enhanced. This is not evident from the image shown in Figure 6C. One would rather conclude that glial cells are terribly swollen. Moreover, endogenous labeling of Rab5 should be performed.

Expression of a leak K^+^ channel suppresses the zydeco induced seizure phenotype. This could affect cell swelling and thus seizure suppression might not be due to K^+^ buffering. Did the authors test whether increase of Na^+^ resulted in a similar suppression?

*Reviewer #3:*

After reading this paper, I realized that I might have made a mistake in accepting the review, because this is out of my field and I don't feel competent to offer a detailed critique of the data in the figures. I find the model to be very interesting, however, and the paper is quite impressive, containing data supporting a detailed model for the mechanisms involved in seizures induced by dysfunction of the interactions between cortex glia and neuronal soma. The one missing piece is that they don't show that the candidate K^+^ channel, Sand, is in fact abnormally endocytosed when CN activity is increased as a consequence of Ca^2+^ elevation.

I don't understand the idea that pan-glial *zyd* knockdown results in lethality "from the requirement of *zyd* in glial subtypes other than cortex glia", as distinguished from the idea that the *zyd* mutation is not a null while RNAi may provide a close-to-null phenotype. These are really the same idea, since in a *zyd* mutant the protein would be lost from all classes of glia. It must be the case that the zyd mutation is a hypomorph. It would be nice to have a CRISPR-induced *zyd* null to confirm that *zyd* is indeed a lethal and rule out off-target effects of the RNAi.

"less seizures" should be replaced with "fewer seizures". This error is made throughout the paper.

Subsection “Cortex glial calcineurin activity is required for seizures in zyd mutants”, last paragraph: Is it known whether any CN A subunit can associate with any B subunit? If so, this should be mentioned. In particular, is it known that 14D and 14F can both associate with CanB? Can they associate with CanB2? If they can associate with CanB2, is CanB2 not expressed in cortex glia?

Subsection “Pharmacological targeting of the glial calcineurin pathway rescues zyd seizures”: Do the effects of CsA all go through cyclophilins? There are 3 in flies, from looking at Flybase. Would it be of interest to knock these down to see if this eliminates the effect of CsA on seizures?

The most obvious missing experiment in this paper is that they don't test the hypothesis that Sand is abnormally endocytosed in *zyd* mutants, and that this is the cause of the phenotype. They could address this by making a tagged version of Sand and determining if its distribution within cortex glia (surface vs. endosomes) changes in *zyd* mutants. This seems like an important experiment to me but I recognize that: 1) if they haven't started it, this would take 6 months or more to complete, thus delaying the paper; 2) this paper already contains a lot of data.

Figure 7: To clarify the diagram, they might wish to indicate more clearly that endocytosis is depleting the K^+^ channels at the surface. The symbols (yellow rectangles) in the endosomes could be labeled as "Sand (and other) K^+^ buffering agents)".

The Discussion is very long (~2400 words) and I imagine few people will read the entire text. They should consider reducing its length to cover only the essential aspects.

---

## [Author Response]

[Editors’ note: the author responses to the first round of peer review follow.]

Reviewer #1:[…] 1) Physiological data traces are shown as representative throughout the paper, without quantification. Thus, it is difficult to assess phenotype variance in wild type and mutations. It should be straightforward to quantify power of burst frequencies or some such measure and report for wild type and mutant in at least a few of these figures (Figure 2C, Figure 5C, Figure 7D).

We thank the reviewer for this suggestion and have now included quantification of the physiological data in Figures 2C, 5C and 8D.

2) The data sets generally are presented without wild type or genotype controls. These need not be added to every figure panel, but greater presentation of wild type data is important. This is essential in Figure 1A and Figure 7D at a minimum, but would be nice to see elsewhere as well.

We have now added control data to the figures as suggested.

3) Video of the seizure phenotype would be helpful. Most quantification is of 'percent flies with seizures'. This is an experimenter-based metric and seems solid given that the phenotypes are generally 100%. But, greater confidence in the data would be generated by showing how robust and obvious the phenotype actually is to the naïve observer.

We have now included representative videos of the heat-shock induced seizure phenotype for wildtype, *zydeco, zydeco*/CanB2^RNAi^, sand^RNAi^ and *zydeco*/Rab5^RNAi^ (Videos 1, 4, 5, 6, 9 and 11 to directly show the behavior).

4) The weakest point of the data set is the connection to endocytosis of 2P channel. Ideally, the channel would be monitored and shown to have altered trafficking.This is likely impossible given the current tools. However, examination of Rab5 itself could be indicative of many different cell biological changes. I am not sure what to suggest. It might be sufficient to remove these data and simply cite the two possible mechanisms of action for the signaling system rather than presenting a strong conclusion at this point. There is sufficient data in the manuscript without this experiment. On the other hand, genetic interaction with Hrs was documented in the secondary screen and Hrs is a late endosomal protein. An alternative to emphasize the endosomal interactions, with addition of new data, without proposing a direct mechanism of action.

Indeed, this was a key experiment we were doing, but did not yet have the data for the initial submission. We have now generated transgenic strains expressing GFP-tagged sand to allow direct monitoring of the protein’s localization in wildtype vs. *zydeco* cortex glia. Using these lines, we were able to directly visualize altered localization of sand in *zydeco* mutants. These data provide key support linking elevated baseline Ca^2+^ and calcineurin activity driving enhanced endocytosis in *zydeco* cortex glia with a disruption in sand surface expression and K^+^ buffering. We have included these data in the revised manuscript (data in Figures 6, described in the subsections “Sand is not differentially phosphorylated in zyd cortex glia” and “Enhanced endocytosis in zyd cortex glia leads to reduction of plasma membrane SAND”). Together with the conditional inhibition of endocytosis experiment using shibirie^ts^ (Figure 7E-G) and the increased Rab5 puncta, we feel this new data adds strong support to our genetic analysis. Together with the reversal of zydeco seizures by EKO expression, our data include multiple approaches that support the current model.

5) Please provide primary (sample data) for calcium transients in Figure 2 and 5.

We have now included sample data for Ca^2+^ imaging in wildtype, zydeco, CanB2^RNAi^, zydeco/CanB2^RNAi^ and sand^RNAi^ in the data (Videos 2, 3, 7, 8 and 10).

6) The data in Figure 5 presents the phenotypic similarity of zyd and sand mutants. The authors conclude that they are part of the same process. The problem is that both mutant conditions are fully penetrant, so putting them together is fairly inconclusive. The authors argue that seizures do not get worse in the double mutant condition, but there does not seem to be any quantitative way to assess a seizure getting worse, and the frequency is maxed out already. The authors could text a double heterozygous condition. They authors could text the double mutants and look for changes in threshold or over-express sand in the zyd mutant background. Something along these lines should be performed to strengthen the desired conclusion.

To assess whether the seizure phenotype is enhanced in zyd/sand^RNAi^ combinations, we assayed several seizure parameters, including seizure temperature threshold (Figure 5B), seizure kinetics (Figure 5—figure supplement 1B) and basal room temperature behaviors using different zyd/sand^RNAi^ combinations. We now show that while heterozygote *zyd* females (zyd/+) do not seize, and only 50% of animals expressing 2 copies of sand^RNAi^ seize, combining these two elements does not enhance the seizure phenotype as expected if the two genes are acting in the same pathway (Figure 5—figure supplement 1B). We have added these data to the revised manuscript (Figures 5B, Figure 5—figure supplement 1B).

7) In Figure 1D, the authors present recordings in the giant fiber system, which are presumed to be recordings from muscle output, as is standard in the field (though not specified in the text). The authors refer to a seizure, but do not define what level of change in activity represents a 'seizure'. Second, and just as important, they refer to this change as indicative of altered intrinsic excitability. This conclusion is not appropriate without directly recording from central neurons since intrinsic excitability has a precise definition. Better definition of criteria and an different interpretation, more closely tied to the date, are necessary here.

We agree the classical giant fiber assay for seizure threshold induction is a crude one – injecting current into the head to induce a sustained burst in giant fiber activity that is recorded as continuous seizures in the DLM flight muscles. We have now downplayed this data in the text, as it is not a key point for the current data set. However, the assay has been used in the field to show that numerous bang-sensitive mutants that alter neuronal proteins lead to a reduction in that threshold voltage. To our knowledge, *zydeco* is the only bang-sensitive mutant that does not. We now specify in the text that the output of the giant fiber system is recording muscle output, as is standard in the field, and added a more detailed seizure definition to the Materials and methods section.

Reviewer #2:[…] Overall the paper would greatly benefit from shortening.

We have greatly shortened the manuscript as suggested by the reviewer, while trying to preserve the key experimental evidence.

zydeco mutants are homozygous viable, but suppression of zydeco expression by RNAi results in lethality. To me this suggests that RNAi is not the best way to assay this gene and therefore other approaches should be used to discriminate between a developmental or a physiological role of zydeco (e.g. conditional rescue in cortex glia instead of conditional RNAi based knockdown).

The differences between zyd^RNAi^ knockdown and the *zyd* mutant likely arise from the *zyd^1^* mutation being a hypomorphic allele rather than a complete null – indeed we isolated three *zydeco* mutants in our initial screen – two point mutations in the pore of the exchanger, and one stop codon that deletes the last part of the extreme C-terminus of the protein. We have shown that conditional rescue with ZYD only in adult flies rescues the phenotype (Melom and Littleton, 2013). In the current study, conditional knockdown of ZYD in adult flies recapitulates the *zydeco* mutant phenotype. Together, these results suggest that *zyd^1^* is likely a hypomorphic mutation and that the functional expression of ZYD in cortex glia is required to prevent seizures. We have now edited the text to make it clearer (subsection “Mutations in a cortex glial NCKX generate stress-induced seizures without affecting brain structure or baseline neuronal function”, third paragraph). For most all of our data analysis, we use the *zydeco* mutant rather than the RNAi.

The authors state that the zydeco seizure phenotype is not due to morphological abnormalities of the cortex glia and show a rather low-resolution confocal image. Use single cell analysis (MARCM, Brainbow) and TEM.

We have now included high resolution images (Figure 1A) and quantification of the data (cortex volume occupied by cortex glial processes and cortex glial cell body volume – Figure 1B, C) as well as EM (Figure 1D) as suggested by the reviewer.

The suppressor / enhancer screen is interesting, but the authors should present all gene names. For all positive candidates, at least one additional RNAi line or better a mutant should be used. This is mentioned in the text, but I saw no information on the results. Please include this in the table as well (e.g. how many of the hits were independently verified?).

Supplementary file 2 provides the full list of genes (CG#s and gene names) that were knocked down in the screen. For the top hits from the initial screen, we tested other RNAi as well as mutant alleles when available, and performed a secondary screen to test the glial sub population in which the manipulated gene is important. This is now better described in the text and in Supplementary file 2. In addition, we now refocus on our primary hit -- the calcineurin pathway and have removed the discussion of the other modifiers, which will require further analysis in a future study.

The screen identified that knockdown of the phosphatase Calcineurin B2 rescued stress induced seizures in zydeco mutants. A similar rescue was observed using three additional non-overlapping RNAi constructs. However, according to flybase the RNAi constructs used in this experiment are indeed overlapping (and have quite some sequence homology to CanB). A mutant analysis should be done!This is also true for CanA-14F and Pp2B14D, where mutants and P-insertions are available.

We assayed knockout alleles of the CN gene family (CanB, CanB2, CanA-14F and Pp2B14D, Author response image 1). The results for CanB2^KO^ are now included in the manuscript (Figure 2A). However, as for individual CanA RNAi knockdowns, combining individual knockout alleles for either Pp2B-14D or CanA-14F with *zyd* did not significantly improve the *zyd* phenotype, while combining a single copy of each KO allele (the double Pp2B-14D/CanA-14F knockout is homozygote lethal) slightly improved the seizure phenotype, possibly due to the remaining functional alleles of the two genes. To further confirm the key role of Calcineurin in the zydeco pathology, we used multiple different assays, including gene knockdown, dominant-negative transgenes and direct assays of Calcineurin activity (CalexA). Together, we feel these combined data strongly support a key role for Calcineurin in the zydeco pathology.

Increased Ca^2+^ levels in zydeco mutants result in increased Calcineurin activity. The image provided in Figure 3 does not match the western and quantification data.

CalexA images of the larval VNC were taken at an exposure that did not cause saturation in *zyd*, so the RNAi image looks like there is no signal. We have now replaced these with better representative images and added an enhanced image of CalexA in *zyd*/CanB2^RNAi^ cortex glia (Figure 3A-C, C’). The Western blot was done on adult head lysates, and the slight differences in GFP expression levels are likely the result of the different origin of the tissue tested.

Pharmacological blockage of the Calcineurin pathway rescues stress-induced seizures in zydeco mutants. The statement that this treatment affects the Calcineurin pathway in glial cells is not supported by the data and should be removed.

We have now edited this statement as suggested (subsection “Cortex glial knockdown of the two-pore-domain K^+^ channel, sandman, mimics zyd seizures”).

In a next step the authors study putative Calcineurin target proteins including the K^+^ channel Sandman. Again, the analysis is on the level of RNAi induced knockdown only and should include a mutant study. A crucial aspect of the paper is how Sandman function can be coupled to Ca^2+^ levels controlled by Zydeco. The mammalian Sandman orthologue is regulated by phosphorylation and two of the four phosphorylated tyrosine residues are conserved. Inhibition of kinases possibly phosphorylating Sandman gave negative results which does not allow any conclusions. Specific sandman mutants would help.

We have now generated transgenic *Drosophila* strains expressing GFP-tagged sand using the UAS/gal4 system. We use this line to show that the expression level of sand:GFP is reduced when co-expressed with sand^RNAi^ (Figure 6A-B). Unfortunately, no mutant is available, and a second RNAi for sand was found to be lethal when expressed either with a pan-glial or cortex glial specific drivers. We also used our transgenic GFP-labeled lines to directly monitor the phosphorylation status of sand, and found that the protein is not differentially phosphorylated in *zyd* relative to wildtype cortex glia (though it is phosphorylated in both genotypes, Figure 6C). These new data are now included in the manuscript (subsection “Sand is not differentially phosphorylated in *zyd* cortex glia”).

Throughout the text the authors frequently compare *Drosophila* cortex glia with mammalian glia – mostly astrocytes. This is in particular relevant in the context of K^+^ buffering. However, insect neurons are generally unipolar which means that the neuronal cell body is not involved in signal propagation as it is in mammalian neurons. The question that needs to be addressed is therefore where the different exchangers / channels (Zydeco or Sandman) are expressed within the cell.

We previously generated antisera to the ZYD protein and demonstrated that it localizes to the cortex glial membrane wrapping that surrounds neuronal somas – knockdown of zyd with RNAi confirmed the specificity of the antisera (Melom and Littleton, 2013). There are no antibodies available to the Sand K_2P_ channel, but our expression of the Sand::GFP transgenic protein shows that it traffics to the cortex glial plasma membrane as well, suggesting both the exchanger and channel are likely to be functioning at the plasma membrane as expected. Although the precise localization of the AIS action potential trigger zone is poorly defined in *Drosophila* neurons, we would expect these excitability changes originating from the soma to significantly influence action potential generation.

The authors claim that endocytosis is enhanced. This is not evident from the image shown in Figure 6C. One would rather conclude that glial cells are terribly swollen. Moreover, endogenous labeling of Rab5 should be performed.

We initially tried to use available endogenous tagged versions of Rab5. However, since the tagged Rab5 is also expressed in neuronal cell bodies, the images are difficult to analyze and interpret. The volume measurements of cortex glial processes (Figure 1B) and cell body (Figure 1C) that we have now included in the manuscript demonstrate that there is no change in cortex glial cell volume. Together, the increase in Rab5 positive puncta presented in Figure 7C-D, the reduced plasma membrane expression of sand:GFP (Figure 6) and the rescue effect observed when endocytosis is conditionally inhibited by using Shi^ts^ (Figure 6E-G) support enhanced endocytosis as the primary mechanism.

Expression of a leak K^+^ channel suppresses the zydeco induced seizure phenotype. This could affect cell swelling and thus seizure suppression might not be due to K^+^ buffering. Did the authors test whether increase of Na^+^ resulted in a similar suppression?

We now show that sand localization is altered in *zyd* cortex glia (Figure 6) and that there are no significant changes in cortex glial cell body volume between wildtype and *zyd* mutants (Figure 1B-C), suggesting the cause of *zyd* seizures is the alteration in ionic homeostasis rather than cell swelling. The rescue we observe when EKO is overexpressed in cortex glia supports this model. We have not tried to manipulate external Na^+^, as all our experiments are performed in live undissected animals. However, knocking down genes that are involved in Na^+^ homeostasis did not modify the *zyd* phenotype or cause obvious behavioral phenotypes on a wildtype background (Supplementary file 2).

Reviewer #3:After reading this paper, I realized that I might have made a mistake in accepting the review, because this is out of my field and I don't feel competent to offer a detailed critique of the data in the figures. I find the model to be very interesting, however, and the paper is quite impressive, containing data supporting a detailed model for the mechanisms involved in seizures induced by dysfunction of the interactions between cortex glia and neuronal soma.The one missing piece is that they don't show that the candidate K^+^ channel, Sand, is in fact abnormally endocytosed when CN activity is increased as a consequence of Ca^2+^ elevation.

Indeed, this was a key experiment. We now shown that GFP-tagged sand cortex glial membrane localization is reduced in *zyd* relative to wildtype (Figure 6). This provides key support linking elevated baseline Ca^2+^, enhanced calcineurin activity and enhanced endocytosis in *zydeco* cortex glia with a disruption in sand surface expression and K^+^ buffering.

I don't understand the idea that pan-glial zyd knockdown results in lethality "from the requirement of zyd in glial subtypes other than cortex glia", as distinguished from the idea that the zyd mutation is not a null while RNAi may provide a close-to-null phenotype. These are really the same idea, since in a zyd mutant the protein would be lost from all classes of glia. It must be the case that the zyd mutation is a hypomorph. It would be nice to have a CRISPR-induced zyd null to confirm that zyd is indeed a lethal and rule out off-target effects of the RNAi.

Indeed, the differences between *zyd* RNAi knockdown and *zyd* mutants likely arise from the *zyd* mutation being a hypomorph allele rather than a complete null as discussed above. We have now revised this point in the text and have removed the *zyd* RNAi lethality data, as it does not add to the current analysis. We tried to generate a CRISPR null for *zyd*, however, the gene maps to 20C in X chromosome heterochromatin, making the locus unfeasible for manipulation due to highly repetitive sequences in this region. In the current study, we chose to focus on the pathway that is mis-regulated downstream of ZYD. We plan to better characterize zyd function (both the wildtype and the mutated forms) in future studies.

"less seizures" should be replaced with "fewer seizures". This error is made throughout the paper.

Thanks for catching this – we have changed it throughout the text.

Subsection “Cortex glial calcineurin activity is required for seizures in zyd mutants”, last paragraph: Is it known whether any CN A subunit can associate with any B subunit? If so, this should be mentioned. In particular, is it known that 14D and 14F can both associate with CanB? Can they associate with CanB2? If they can associate with CanB2, is CanB2 not expressed in cortex glia?

Several previous studies describe co-activity of CanB2 and Pp2B-14D (PMIDs: 25081566, 20561515, 12019233), and a study by Takeo et al. (PMID 22421435) showed that Pp2B-14D, CanA-14F, CanB2 and Calmodulin can all be coimmunoprecipitated, and that *Pp2B-14D* and *CanA-14F* show redundant function. Knockdown of the second *Drosophila* CanB subunit or a knock-out allele for it, had no effect on *zyd* phenotype, suggesting CanB2 is the primary CanB subunit that functions within cortex glia.

**Author response image 2. respfig2:** 

Subsection “Pharmacological targeting of the glial calcineurin pathway rescues zyd seizures”: Do the effects of CsA all go through cyclophilins? There are 3 in flies, from looking at Flybase. Would it be of interest to knock these down to see if this eliminates the effect of CsA on seizures?

There are multiple reports regarding additional potential targets for CsA, as well as for other CN inhibitors. While the main effect of CsA is to inhibit the calcineurin pathway by forming a complex with the immunophilin Cyclophilin, FK506 is proposed to block the activation of calcineurin through the formation of complexes with a different immunophilin called FK506 binding protein (FKBP) 12. We found that application of FK506 can also partially rescue *zyd* seizures (Figure 4D), suggesting complex regulation in CN activity. In addition, knocking down all 3 of the *Drosophila* cyclophilins, as well as the *Drosophila* homologue of FKBP12 did not alter the *zyd* phenotype or cause an obvious behavioral phenotype when knocked down on a wildtype background (Supplementary file 2). We primarily use the pharmacological approach to simply support our much stronger genetic interactions.

The most obvious missing experiment in this paper is that they don't test the hypothesis that Sand is abnormally endocytosed in zyd mutants, and that this is the cause of the phenotype. They could address this by making a tagged version of Sand and determining if its distribution within cortex glia (surface vs. endosomes) changes in zyd mutants. This seems like an important experiment to me but I recognize that: 1) if they haven't started it, this would take 6 months or more to complete, thus delaying the paper; 2) this paper already contains a lot of data.

As noted above, we have generated transgenic strains expressing GFP-tagged sand to allow direct monitoring of the protein’s localization in wildtype versus *zydeco* cortex glia. Using these lines, we were able to directly visualize altered localization of the protein in cortex glia of *zyd* mutants. This provides key support linking elevated baseline Ca^2+^, enhanced calcineurin activity and enhanced endocytosis in *zydeco* cortex glia with to a disruption in sandman surface expression and K^+^ buffering. We have now included these data in the revised manuscript (Figure 6).

Figure 7: To clarify the diagram, they might wish to indicate more clearly that endocytosis is depleting the K^+^ channels at the surface. The symbols (yellow rectangles) in the endosomes could be labeled as "Sand (and other) K^+^ buffering agents)".

The model in Figure 8A now states more clearly that the enhanced endocytosis leads to altered surface expression of sand.

The Discussion is very long (~2400 words) and I imagine few people will read the entire text. They should consider reducing its length to cover only the essential aspects.

We have now substantially shortened the Discussion to cover only the essential points to highlight the importance of this work.